# Integrating pheromonal and spatial information in the amygdalo-hippocampal network

María Villafranca-Faus [1], Manuel Esteban Vila-Martín[1,2], Daniel Esteve [3], Esteban Merino [1], Anna Teruel-Sanchis [1,2], Ana Cervera-Ferri [1], Joana Martínez-Ricós [1], Ana Lloret[3], Enrique Lanuza [1,2,4✉] & Vicent Teruel-Martí [1,4✉]

Vomeronasal information is critical in mice for territorial behavior. Consequently, learning the territorial spatial structure should incorporate the vomeronasal signals indicating individual identity into the hippocampal cognitive map. In this work we show in mice that navigating a virtual environment induces synchronic activity, with causality in both directionalities, between the vomeronasal amygdala and the dorsal CA1 of the hippocampus in the theta frequency range. The detection of urine stimuli induces synaptic plasticity in the vomeronasal pathway and the dorsal hippocampus, even in animals with experimentally induced anosmia. In the dorsal hippocampus, this plasticity is associated with the overexpression of pAKT and pGSK3β. An amygdalo-entorhino-hippocampal circuit likely underlies this effect of pheromonal information on hippocampal learning. This circuit likely constitutes the neural substrate of territorial behavior in mice, and it allows the integration of social and spatial information.

[1] Neuronal Circuits Laboratory, Dept. of Anatomy and Human Embriology, Faculty of Medicine, University de València, Valencia, Spain. [2] Laboratori de Neuranatomia Funcional, Dept. de Biologia Cel·lular, Fac. CC. Biològiques, Universitat de València, Valencia, Spain. [3] Department of Physiology, Faculty of Medicine, University of Valencia, Health Research Institute INCLIVA, CIBERFES, Valencia, Spain. [4] These authors contributed equally: Enrique Lanuza, Vicent Teruel-Martí. ✉email: enrique.lanuza@uv.es; vicent.teruel@uv.es

Mice are territorial animals[1] that use urine marks to delimitate the boundaries of their territories[2]. Male mice defend their territories with aggressive behavior against intruders, and dominant males possess larger territories with richer resources[3]. Females recognize the owner of each territory by the detection of urine marks and choose which territories they visit to find potential mates. The recognition of individual animals is based on the pattern of major urinary proteins present in urine spots[4], and the detection of urinary proteins depends on the vomeronasal system[5]. Therefore, both males and females need to generate a spatial map, presumably integrated in the hippocampus-dependent memory, in which the vomeronasal signals encoding the identity of the territory owners (the "who" component of memory) should be key features, together with other sensory cues. Consequently, hippocampal activity should be strongly influenced by the vomeronasal system. However, to our knowledge, this sensory modality has not been taken into account in previous studies on the role of the hippocampus in rodents[6]. In addition, the boundaries of the territory defended by an individual are unstable, requiring continuous learning of the modifications. Thus, the vomeronasal input to the hippocampus should show synaptic plasticity allowing to update the territorial information.

In fact, male-specific major urinary proteins induce spatial learning[7], although the specific role of the hippocampus remains to be shown. To investigate whether vomeronasal information influences learning-related activity in the hippocampus, we have recorded the simultaneous local field potentials (LFP) in the vomeronasal cortical amygdala and dorsal CA1 (dCA1) in awake, head-fixed animals running in a virtual reality system where urinary stimuli were presented associated with a particular context (Fig. 1). In the vomeronasal amygdala, activity was recorded in the posteromedial cortical nucleus (PMCo), which can be considered the vomeronasal primary cortex[8]. In parallel, in anesthetized animals we have tested the effects of high-frequency stimulation (HFS) of the vomeronasal pathway in the induction

of long-term potentiation (LTP) in the PMCo and, at the same time, in the Schaffer's collaterals linking the hippocampal CA3 with CA1. To investigate whether this synaptic plasticity can be induced by vomeronasal signals contained in urine, we have also tested the induction of LTP in the PMCo and CA1 applying male urine in the nostrils of female mice, including animals with experimental anosmia induced by irrigation of the nasal mucosa with zinc sulfate. In addition, in freely behaving female mice allowed to investigate male urine we have investigated whether the exposure to urinary signals results in an increased expression of proteins involved in synaptic plasticity in the amygdaloid PMCo and dorsal hippocampus.

Since the PMCo does not project to the dorsal hippocampus[8], we performed tract-tracing studies that revealed an indirect pathway that could mediate the access of vomeronasal information to the hippocampus via the lateral entorhinal cortex (LEnt). To confirm the activation of this pathway by male pheromones, the c-fos expression in the amygdaloid PMCo, LEnt, and dorsal CA1 were analyzed in females, following the exposure to male chemical signals.

## Results

The primary vomeronasal cortex is the PMCo[8] and, therefore, it may be the first step of a putative pathway for integrating pheromonal signals into the hippocampus-dependent memory. Thus, simultaneous LFP in PMCo and dCA1 were recorded in awake head-fixed mice navigating a virtual environment associated to visual, olfactory, and vomeronasal stimuli in different virtual contexts. The system consisted of one training (c1) and three testing corridors (c2–c4), where mice can navigate using a running wheel in a one-dimensional space, forcing the exploration of experimentally controlled stimuli (Fig. 1a, b and Supplementary Movie 1). Virtual corridors were divided in four sectors (s1–s4) characterized by a different pattern of visual cues in the walls, allowing the experimental animals to generate a representation of

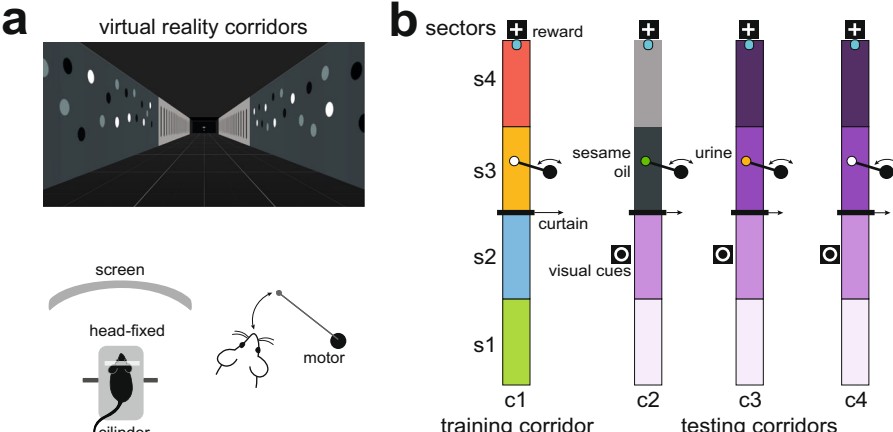

**Fig. 1 Behavioral setup: configuration based on virtual environments. a** Subjects were head-fixed to metal bars, in turn connected to a stereotaxic frame. Movement was only allowed in a single dimension through a low-friction Styrofoam cylinder placed in front of a curved TFT screen. The movements of the cylinder were recorded by an infrared sensor and converted into virtual reality movements (Arduino Mega board, 2560 Rev3), thus simulating navigation. The self-developed virtual environments were implemented in Unity3D software (Unity Technologies, 2018). **b** Drawing of the virtual corridors, viewed from above. The virtual corridors were designed with four distinct track patterns on the inner walls, which differed along the longitudinal line. The wall patterns visually define four sectors per corridor, s1–s4, with different profiles between training and test corridors; an intermediate wall (curtain) separates the corridor into two compartments, containing sectors s1 and s2, and s3 and s4, respectively. The walls were designed with colors within the range of wavelengths perceived by the mouse visual system[49,50]. A final reward zone was identified with a distinctive signal (white cross), where a 1% sucrose drop is provided. In sector 3 of each corridor, a cotton swab attached to a silent motor is slowly advanced to the nose. The swab is moved until it is close enough to the nose to allow contact with the stimuli. In the training corridor at s3, a clean cotton swab is presented, while in the test corridors the cotton is impregnated with different chemosensory cues: a non-pheromonal olfactory cue (sesame seed oil) at c2/s3, male urine at c3/s3 and only a clean cotton swab at c4/s4.

different spatial contexts. In s2, salient visual cues (additional to those in the walls) were present, whereas in s3 different chemosensory stimuli were present in each corridor: a non-pheromonal olfactory stimulus (c2, sesame oil), a vomeronasal stimulus (c3, male urine), and a clean cotton swab in c4.

**Coupled theta activity was present in PMCo and CA1 during virtual navigation.** During navigation high levels of theta activity (5–12 Hz) are present in the hippocampal LFP. In our experiments, this distinct profile of activity was accompanied with a similar pattern in the PMCo (Fig. 2a), suggesting an amygdalo–hippocampal dynamics correlated with exploratory behaviors. Accordingly, spectral coherence between both signals demonstrated a peak in theta oscillations (Fig. 2b, top; representative case in 2c). The grand average of the coherence showed a predominant peak in the theta range (Fig. 2e). We further explored the directionality of this connectivity by performing Granger causality analyses. We computed the causal measures over time windows of 1 s to assess the flow of information between both recording areas. Granger analysis revealed a predominant causal state with CA1 leading PMCo, and a less prevalent causality where PMCo drives hippocampal oscillations. In a substantial number of segments, both directionalities were present, without being able to establish a dominant causality (Fig. 2b, d). The distribution of peaks of causality detected over the theta range for all cases displayed differences between the predominant frequency of the significant causal windows (PMCo → CA1, $f = 7.57$ Hz, CA1 → PMCo, $f = 7.07$ Hz; $t_{140} = 3.83$, $p = 0.00019$; Fig. 2f), consistent with high-coherence epochs (Fig. 2b, f). When we measured the differential proportion of detected causal epochs between regions of the virtual environments (Fig. 2g), we observed only a general significant effect in the change between sectors ($F_{3,73} = 3.71$, $p = 0.015$), with the increase of ratio with directionality PMCo → CA1 in the transition from s2 to s3 (s2–s3, $t_{73} = 2.81$, $p = 0.031$). These results could be an evidence of a differential profile of causal activity in the theta oscillations of both structures between visual cues (s2) and chemosensory stimuli (s3).

In summary, during exploratory states oscillatory theta activity in PMCo and CA1 coexists. In this dynamics, hippocampal theta seems to coordinate a common activity in both areas, with information transfer between them.

**A common pattern of theta-nested gamma activity in the amygdalo–hippocampal network is induced by chemical signals.** A distinguishing feature of this theta profile was its cooccurrence with gamma episodes (30–200 Hz; Fig. 3a), a well-defined cross-frequency interaction consistent with phase coding related to the formation of hippocampal spatial maps[9]. We focused our analysis on the different spectral signatures profiled by gamma waves embedded in single theta cycles (theta-nested spectral components, tSCs). Following Lopes-dos-Santos et al.[10], we detected a significant set of five tSCs (tSC1–tSC5, Fig. 3a, left) across recording experiments and animals, according to previous findings[10,11]. These activity patterns were present in both PMCo and CA1, again suggesting coupled neural processing in both areas.

Along the virtual corridors, different tSC profiles were elicited at both recording sites (Fig. 3b). Such diversity was delimited by hierarchical clustering analysis, in which we used as defining variables the grand average strengths (see "Methods") of each theta component in each context (Fig. 3c). The unsupervised classification system of the functional tSC patterns in the virtual corridors, as a representation of the theta-nested gamma activity, led to a distribution of activities represented in distinguishable

motifs. Thus, similar neuronal activities were observed in delimited clusters according to the Euclidean distance ($E_d$) between them. A cophenetic correlation of 0.85 (day 1) indicated that the cluster tree accurately reflected the original dissimilarity matrix. A 1000 iteration bootstrap procedure validated these findings, showing that PMCo and CA1 motifs were clustered together at c3/s3 in 96% of bootstraps, where the urine stimulus was present, with a predominance of tSC5 (day 1; Fig. 3c, left). On consecutive days, the tSC5-dominated profile was maintained on PMCo and CA1 in response to the urine stimulus (day 2; 86% of bootstrap; Fig. 3c, middle), and only CA1 activity shifted to a predominance of tSC4 (day 3, Fig. 3c, right) with new exposures.

The comparison of tSC patterns between environments with visual and chemosensory cues (Fig. 3d, top), evidenced that the presence of new visual stimuli (c2/s2) elicited different motifs in the PMCo and CA1 ($E_d = 3.46$). The CA1 pattern was characterized by strong tSC3 and tSC4 in CA1, whereas the predominant component of the PMCo pattern was tSC2. A non-pheromonal olfactory stimulus (sesame oil, c2/s3) led to relatively similar tSC profile in both structures ($E_d = 1.29$). In this case, the tSC5 component was prevalent, accompanied by a moderately strong tSC4 component.

The presence of urinary stimuli (c3/s3) was capable to induce a shared pattern of tSCs in both areas ($E_d = 0.81$), characterized by a strong tSC5 and a relative low activity in the rest of the gamma frequencies (Fig. 3d, top). In summary, the visually elicited activity patterns in PMCo and CA1 were clearly different from those observed when vomeronasal stimuli were present.

This same pattern was also detected in c4/s3, where the memory of the previous urine stimulus might be present. Interestingly, we also observed how the previous experience led to a change in tSC4 in c4/s3 (day 3, Fig. 3d, right), where only the spatial component drives the amygdalo–hippocampal pathway to a new theta coding.

Next, we focused on how the different tSC patterns change within a neural structure with navigation. In CA1, the presence of salient visual cues (c2/s2) induced a dominance of tSC3 in comparison with the high strength of tSC5 in c2/s3 with sesame oil stimulus ($E_d = 3.53$; Fig. 3d, bottom). With urine, the difference between visual and chemosensory patterns was similar ($E_d = 3.69$). A minimal distance was found between sesame and urine stimuli ($E_d = 1.62$). PMCo presented similar values of cluster distance (visual-sesame oil, $E_d = 3.63$; visual-urine, $E_d = 3.15$; urine-sesame oil, $E_d = 1.12$), but (as described above) the tSC profile induced by visual stimuli was different front that found in CA1.

In summary, this distinguishing theta/fast-gamma signature (predominant tSC5 and relative low strength of the rest of gamma components) seems to define ongoing sensory processing driving the inclusion of vomeronasal neural representations associated with a particular context in the hippocampal circuit.

**Long-term potentiation of the vomeronasal amygdala and the hippocampal CA1 elicited by tetanic stimulation of the accessory olfactory tract.** To test the hypothesis that the vomeronasal system influences the learning-related activity of the hippocampus, we stimulated at high frequency the vomeronasal pathway and checked the induction of LTP in the PMCo and, at the same time, in the Schaffer's collaterals, linking the hippocampal CA3 with CA1.

The PMCo recordings after the tetanic stimulation of the accessory olfactory tract (aot, Supplementary Fig. 1) revealed the induction of LTP, with an increase in the response amplitude of the evoked potential, significantly higher (Friedmann test $\chi^2_2 = 15.50$, $p = 0.014$) than basal measures. At the longer period

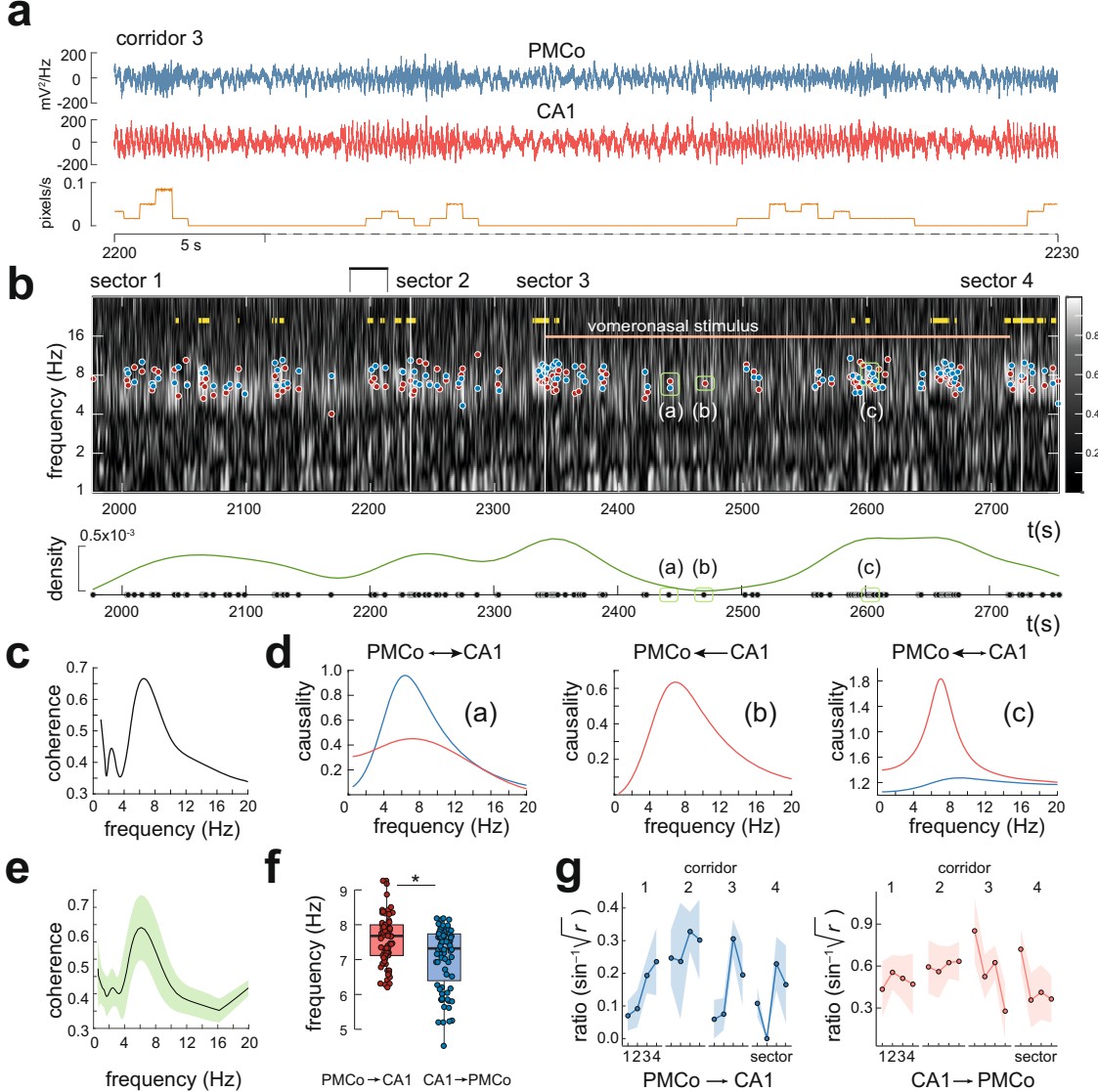

**Fig. 2 Functional relationships of the PMCo–CA1 network in virtual environments. a** Representative case of raw records of local field potentials in PMCo (top) and CA1 (middle) through corridor 3. The movement of the animal is represented in the orange plot (bottom). **b** Wavelet coherogram (top) of the representative case in gray tones, where the lighter ones indicate a high degree of coactivity between both areas. The yellow segments represent motion, and the reddish segment represents the presence of the urinary stimulus. In the diagram, the segment corresponding to the recording represented in a is marked with a black line at the top. On the coherogram, the times (windows of 1 s) in which significant causality was detected in the theta range are marked with dots: blue dots, causality with predominant directionality PMCo → CA1 (a), and red dots for directionality CA1 → PMCo (b, c). The windows of analysis often show points in the two bands, thus demonstrating directionality in both directions, although one of them may predominate over the other. A plot of the density of causal dots along corridor 3 is shown at the bottom. **c** Left: average spectral coherence of the segment shown in **a**, which highlights a peak of synchronization in the theta frequency band. **d** Three plots of spectral causality corresponding to points (a), (b), and (c) in **b**. All three plots show causality peaks in the theta frequency range. **e** Grand average of the spectral coherence of all recordings highlighting the predominance synchronization in the theta range. Data are presented as mean values ± SEM. **f** Statistical comparison between the frequency peaks in both directionalities; two-sided $t$ test, *$p = 0.00019$, $n = 140$ independent samples over five independent experiments; box plots are defined in terms of minima and maxima by whiskers, and the center and bounds of box by quartiles ($Q_1$–$Q_3$). **g** Diagram derived from the statistical analysis showing the proportion of times of each causal directionality (blue, directionality PMCo → CA1; red, directionality CA1 → PMCo) for each of the four sectors for each corridor. Data are presented as mean values ± SEM. Source data are provided as a Source data file. The raw data of the LFP recordings of the experiments in the virtual reality setup generated in this study are available in the Zenodo database[51].

of analysis (20–25 min) the post-comparison showed the highest significant potentiation ($p = 0.0027$; Fig. 4a; for complete multiple comparisons, see Supplementary Table 1). This result reveals the existence of plastic changes in the primary vomeronasal cortex induced by the stimulation of the projection from the accessory olfactory bulbs (AOBs).

Concurrently, the electrical activity in CA1 neurons evoked by single pulses in the Schaffer's collaterals was also recorded before and after the tetanic stimulation of the aot, to check whether vomeronasal input to the amygdala produces synaptic changes in the hippocampus. We found an intensification (Friedmann test, $\chi^2_2 = 17.40$, $p = 0.0006$) in the amplitude of the postsynaptic

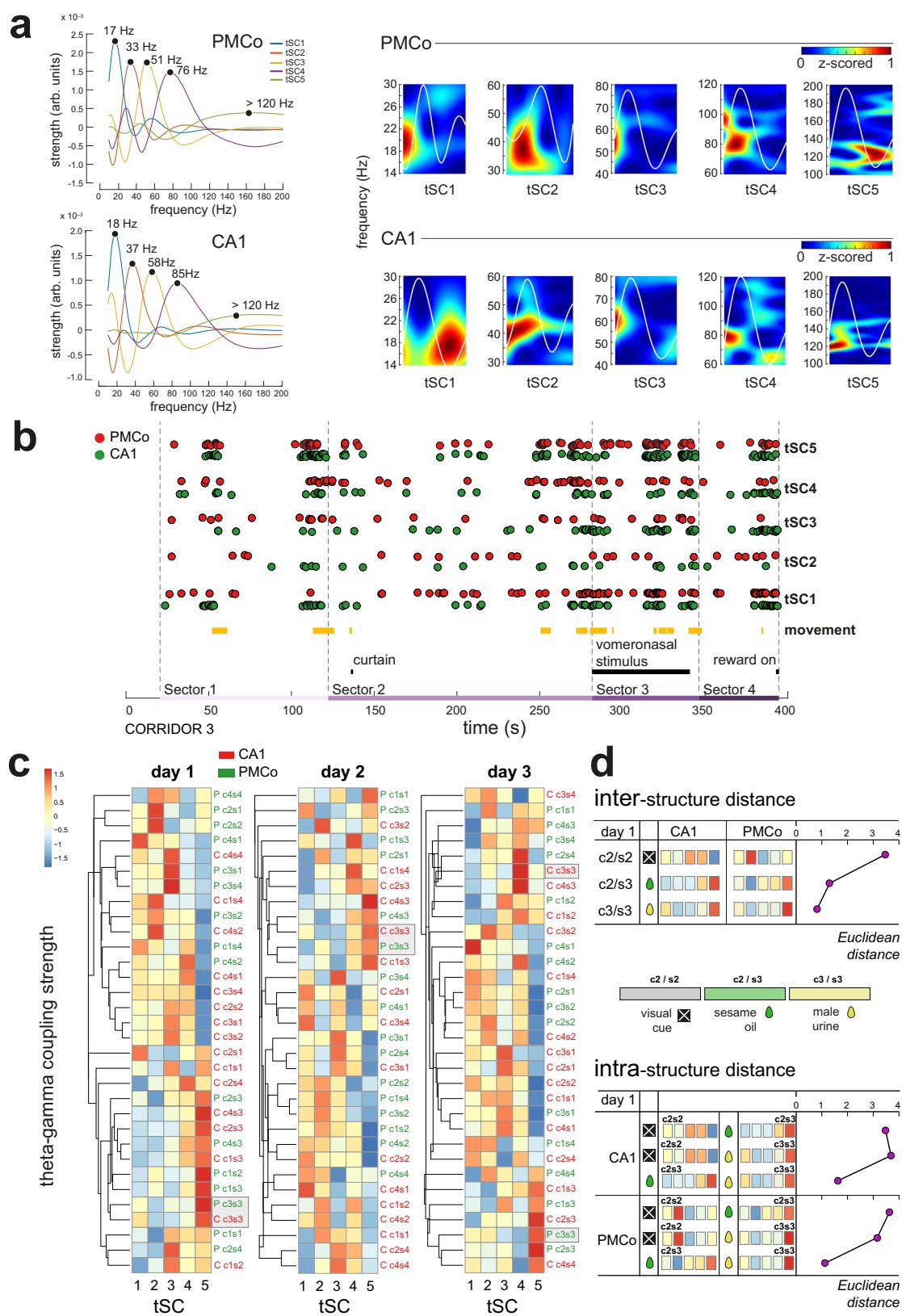

potential in CA1 after tetanic stimulation (Fig. 4b). The 20–25 measure was significantly higher than basal ($p = 0.0001$).

**Exposure to male urine induces synaptic plasticity in the PMCo and the hippocampal CA1 of female mice.** Next, we

investigated whether this synaptic plasticity might be induced in natural conditions by the presence of chemical signals contained in urine. To do so, we tested the induction of LTP in the PMCo and CA1 by the application of male urine in the nostrils of anesthetized female mice, since male urine is an ethologically relevant stimulus with pheromonal activity[4,7]. Thus, we recorded

**Fig. 3 Analysis of the theta-gamma coupling by theta component detection. a** Left: representative theta components (tSCs) extracted from PMCo and CA1 recordings. Spectral signatures in the frequency domain of the defined tSCs are plotted. Right: representative wavelet spectrograms of the five tSCs for both areas of recording. **b** Plot of the detection of the different tSCs in a representative recording case of PMCo (red dots) and CA1 (green dots) over corridor 3. The dots are slightly jittered to facilitate visualization. The diagram also shows the epochs of the animal movement (yellow segments) and the presence of the curtain, vomeronasal stimulus and reward (black segments). **c** Clustering of the grand average of the similarity strength (arbitrary units) of each spectral signature over the tSC, in each corridor, sector, for both recording areas (PMCo, green; CA1, red) for the three recording days. The nomenclature is defined as area (PMCo, P; CA1, C), corridor (c1–c4) and sector (s1–s4). Each row can be understood as a theta-gamma activity pattern correlated with the exploration induced by each of the virtual sectors. Warm colors indicate a strong presence of the tSC in that corridor–sector. The patterns corresponding to P-c3s3 and C-c3s3, i.e., those corresponding to the time when the urine stimulus is present, are framed in gray boxes. Clustering is based on the similarity of the patterns, calculated by Euclidean distance. **d** Plots of the comparison of activity patterns between sectors c2/s2 (visual cues), c2/s3 (olfactory views), and c3/s3 (vomeronasal cues), between both recordings (top) and for the same recording (bottom). The diagram visualizes the Euclidean distance of these comparisons (violet dots). Source data are provided as a Source data file. The raw data of the LFP recordings of the experiments in the virtual reality setup generated in this study are available in the Zenodo database[51].

the amplitude of single pulse-evoked potentials in the PMCo following the stimulation of the aot, and in CA1 after the stimulation of the Schaffer's collaterals, before and 30 min after urine application. The results indicated that male urine elicited LTP both in PMCo (Fig. 4c) and CA1 (Fig. 4d). For both PMCo- and CA1-induced LTP, the statistical analysis revealed a significant amplitude increment compared to the basal response (Friedmann test; PMCo, $\chi_2^2 = 21.22$, $p = 9.48e-5$; Fig. 4c; CA1, $\chi_2^2 = 19.90$, $p = 0.0002$; Fig. 4c). In both cases, post hoc comparisons revealed a significant potentiation in the 20–25 period (PMCo, $p = 0.0001$; CA1, $p = 0.0008$; for complete multiple comparisons, see Supplementary Table 1).

To demonstrate the specific role of the vomeronasal signals contained in urine, we ran the urine-induced LTP experiments in mice previously rendered anosmic by irrigation of the nasal cavity with zinc sulfate. Previous to the introduction of urine, we tested the effect of a neutral olfactory stimulus (citralva). A habituation–dishabituation test (Fig. 5a) confirmed the successful induction of anosmia. In addition, we were able to corroborate in these anosmic animals that the placement of a drop of dye in the nasal cavity diffuses into the lumen of the vomeronasal organ (VNO) and is, at least partially, incorporated into the epithelium (Fig. 5b).

The results of the habituation–dishabituation test (Fig. 5c) showed significant differences between control and lesioned animals throughout the behavioral test ($F_{1,17} = 8.0334$, $p = 0.011$), with exploration time being longer in the control group than in the lesioned one. A post hoc analysis in restricted temporal windows corresponding to the introduction of citralva at 6–8 min showed an increase of exploration in the control group (statistical tendency, $p = 0.072$), which not observed in the zinc sulfate-treated animals ($p = 0.89$). In turn, the introduction of male urine elicited a significant increase in exploration time in control group ($p = 0.034$), but not in lesioned animals ($p = 0.98$). If we restrict the comparison between groups to the 6–8 epoch (citralva) and 12–14 (urine), the analysis demonstrated distinct behavior in response to urine ($p = 0.039$) and a tendency to significant difference in the case of citralva ($p = 0.062$).

Importantly, locomotor activity did not differ between groups, as revealed by the analysis of movement velocity ($t_{17} = 1.42$, $p = 0.17$) and total distance moved ($t_{17} = 1.39$, $p = 0.18$).

LTP experiments on lesioned mice showed an increased amplitude in the postsynaptic responses (Friedmann test, PMCo, $\chi_6^2 = 28.21$, $p = 8.58e-5$; CA1, $\chi_6^2 = 31.78$, $p = 1.80e-5$). The stimulation with citralva did not lead to significant changes in the amplitude of the evoked potentials (20–25 vs. basal: PMCo, $p > 0.99$; CA1, $p > 0.99$). In contrast, the urine stimuli elicited significant increases on the synaptic potentials in both areas (20–25 vs. basal: PMCo, $p = 0.0014$; CA1, $p = 0.0078$), as a sign of LTP (Fig. 5d, e).

**Urine of conspecific males induced a territorial preference and an increased expression of synaptic proteins in the hippocampus.** To address the question of the molecular substrate of urine-induced LTP, we performed a behavioral preference test that temporarily presented to female subjects either male urine, citralva, or saline in a quadrant of the cage. As expected, the presence of male-derived stimuli had a significant effect on exploration time ($F_{2,17} = 9.24$, $p = 0.0019$, Fig. 6a, b). Females exposed to male urine showed significantly longer exploration time in the region of interest (ROI) when compared to animals exposed to saline ($p = 0.0041$) or citralva solution ($p = 0.0048$). However, animals exploring saline or citralva were not significantly different ($p = 0.99$).

We analyzed the expression of proteins related to synaptic plasticity in the amygdala and dorsal hippocampus of these same animals, 90 min after the preference test. We observed significant differences both in the ratio of pAKT/AKT ($F_{2,17} = 6.23$, $p = 0.0107$; Fig. 6c, left; western blots in Fig. 6d) and pGSK3b/GSK3β ($F_{2,17} = 10.71$, $p = 0.0009$; Fig. 6c, right; western blots in Fig. 6d). A significant increase in pAKT/AKT was observed in CA1 with urine presentation when compared to citralva ($p = 0.016$,) and saline groups ($p = 0.037$), which showed nonsignificant differences ($p = 0.85$). In contrast, lower levels of activated GSK3β (higher ratio of pGSK3b/GSK3β) in CA1 were shown in urine-exposed animals than in citralva ($p = 0.0015$) and saline ($p = 0.0038$), which, again, were not different from each other ($p = 0.90$). Molecular results in the amygdala did not show significant differences between groups (see Supplementary Fig. 2).

**Anatomical pathways linking the vomeronasal amygdala and the dorsal hippocampus.** Synaptic plasticity in the pathway from the AOBs to the PMCo suggests that this connection is glutamatergic. To test this possibility, AOB injections of the anterograde tracer tetramethylrhodamine and biotin-conjugated dextranamines (TBDA) were combined with the immunofluorescence detection of the glutamatergic marker VGLUT1. The results confirm the presence of numerous double-labeled terminal boutons in the deep layer I of PMCo (Fig. 7a). To elucidate the connections from the vomeronasal amygdala to the dorsal hippocampus[8,12–14], the retrograde tracer FluoroGold (FG) was injected in the dorsal hippocampus and the anterograde tracer TBDA in the PMCo. The FG injections in the dorsal CA1 did not give rise to reliable retrograde labeling in the vomeronasal nuclei of the amygdala (i.e., the medial amygdala and the PMCo). Although a few retrogradely labeled neurons were found in the medial amygdala, labeling was really very scarce and not present in every case. In addition, we could not confirm the projection from the medial amygdala to the dorsal hippocampus using anterograde tracing[14]. In the same vein, injections of anterograde tracers in the PMCo did not show

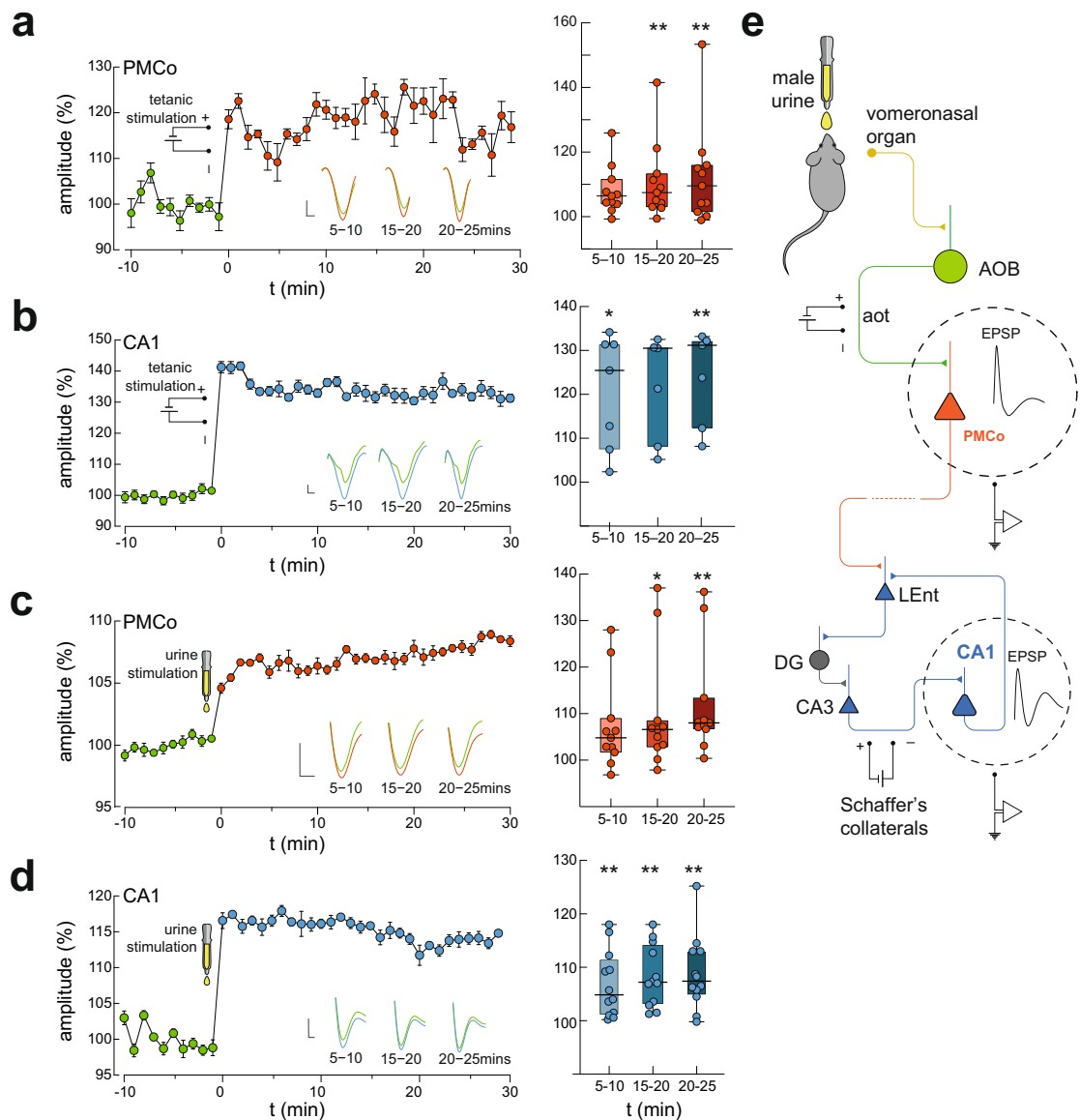

**Fig. 4 Synaptic potentiation at the vomeronasal amygdala and dorsal hippocampal CA1 induced by tetanic stimulation of the accessory olfactory tract and urinary stimuli. a** LTP induction in the PMCo by tetanic stimulation of the accessory olfactory tract (aot; see Supplementary Fig. 1). Left: representative case showing the basal evoked field potentials (green) and the potentiation induced by the tetanic stimulation (red). Each dot represents the average amplitude of six evoked potentials (±SEM). Right: statistical analysis of the differences in amplitude (percentage regarding the mean basal value) of the evoked potentials at 5–10, 15–20, and 20-25 min post-tetanization; $n = 11$ animals. Box plots are defined in terms of minima and maxima by whiskers, and the center and bounds of box by quartiles ($Q_1$–$Q_3$). Calibration: vertical bar, 0.36 mV, horizontal bar, 120 ms. **b** Same schematic representation as in panel a for the LTP induction on the CA1 after the tetanic stimulation of the aot; $n = 7$ animals. Data are presented as mean values ±SEM. Calibration: vertical bar, 0.1 mV, horizontal bar, 100 ms. **c** LTP induction in the PMCo by exposure to conspecific male urine. Left: representative case of LTP induction in PMCo. Right: statistical analysis of the differences in amplitude of the evoked potentials at 5–10, 15–20, and 20-25 min post-stimulation; $n = 11$ animals. Data are presented as mean values ±SEM Calibration: vertical bar, 0.12 mV, horizontal bar, 130 ms. **d** Same schematic representation as in **c** for the LTP induction on the CA1 after exposure to conspecific male urine; $n = 12$ animals. Data are presented as mean values ± SEM. Calibration: vertical bar, 0.9 mV, horizontal bar, 150 ms. For right panels, Friedmann test with Dunn's test for post hoc comparisons, ● statistical tendency with $p < 0.1$, *$p < 0.05$, **$p < 0.01$. Exact $p$ values of Dunn's multiple comparisons are provided in Supplementary Table 1. Horizontal thick lines represent the median values. **e** Schematic diagram of the amygdalo–hippocampal circuit showing the experimental paradigm. Source data are provided as a Source data file.

anterogradely labeled fibers in the dorsal CA1. Thus, we looked for convergent areas where retrogradely labeled cells and anterogradely labeled fibers were simultaneously present. We found a restricted population of retrogradely labeled neurons in layer II of the dorsolateral entorhinal cortex (dLEnt), and the adjacent perirhinal cortex (Fig. 7b, c), in an area where anterogradely labeled fibers with boutons were also present (Fig. 7d). No other area with simultaneous retrograde and anterograde labeling was observed. To

neurochemically characterize the neurons in the dLEnt projecting to the dorsal CA1, two successful injections were processed for immunofluorescence for FG, calbindin, and reelin and stained with DAPI. Retrogradely labeled cells of the LEnt were mainly reelin positive (Fig. 7e). Injections of FG in the dLEnt gave rise to numerous retrogradely labeled cells in the PMCo, especially in the caudal two thirds of this nucleus, and located preferentially in its

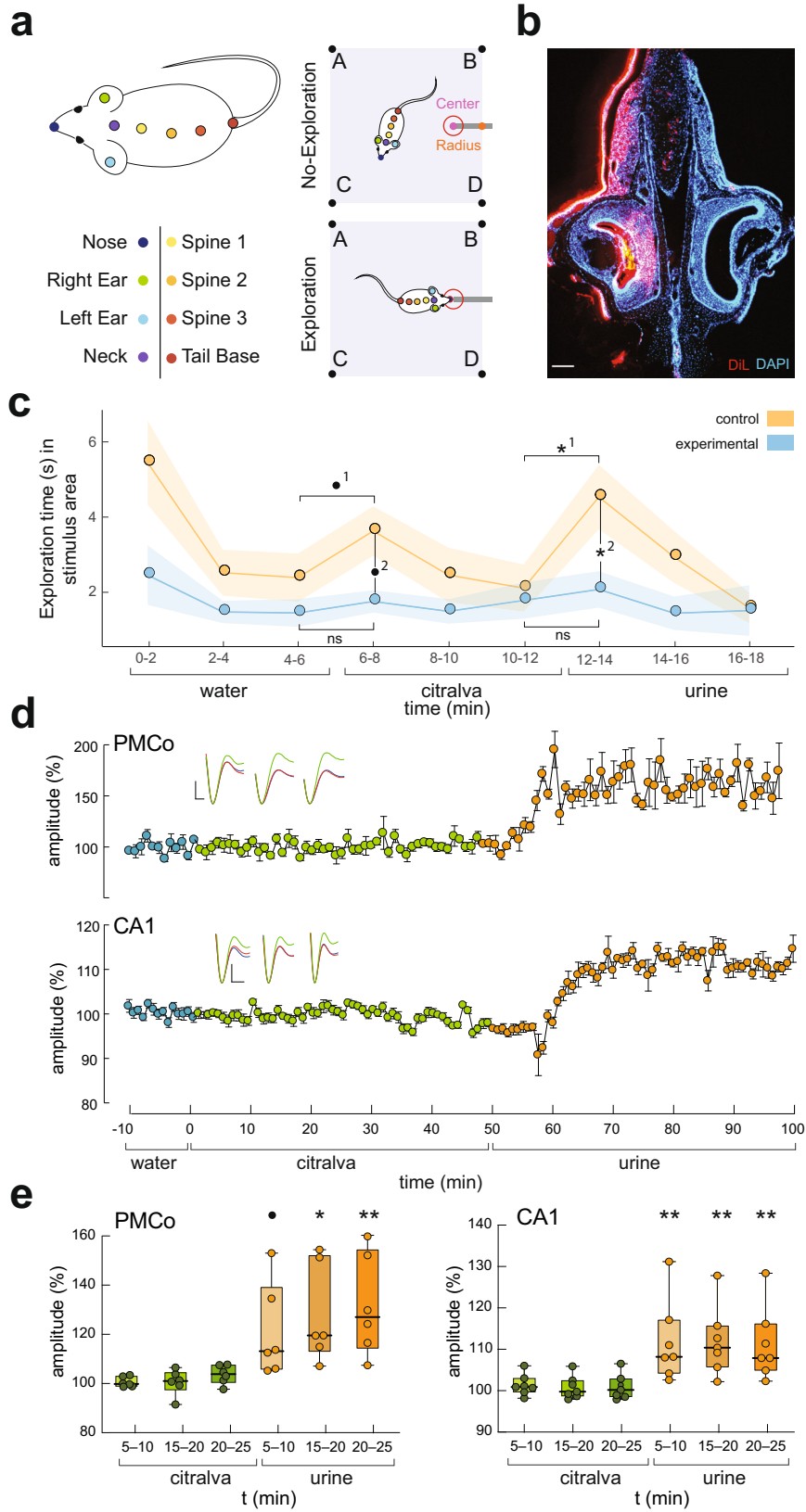

medial aspect (Supplementary Fig. 3), confirming the pathway revealed by the anterograde tracing experiments.

**Male chemical signals induced c-Fos expression in the vomeronasal amygdala and lateral entorhinal cortex.** To

confirm the activation of the PMCo–dLEnt–CA1 pathway by male chemical signals, the c-fos expression in the PMCo, dLEnt and dorsal CA1 was analyzed in female mice exposed to either male urine (experimental group) or a neutral odorant (citralva, control group) presented in a particular area within the test cage. The analysis of the chemoinvestigatory behavior of the animals

**Fig. 5 Potentiation at amygdalo–hippocampal circuit by the vomeronasal components of urinary pheromones in females with lesion of the olfactory epithelium. a** Left: the labels used to train our DLC network both for the mouse and the arena. Exploration was quantified when nose tracking was inside the region of interest, defined as 0.25 times the cotton swab distance. **b** Photomicrograph of a coronal 30-μm-thick section of the vomeronasal organ from a urethane-anesthetized mouse showing DiI Incorporation into the vomeronasal epithelium, counterstained with DAPI. Scale bar **b**, 100 μm. **c** Exploration time (min) in the stimulus area in the habituation–dishabituation test. Results show that, through the test, control mice ($n = 10$) explore the stimulus for longer than the experimental group (olfactory epithelium injured, $n = 9$). The introduction of male urine induced in control mice a significantly higher exploration time than in experimental mice. Repeated-measures ANOVA test, ns $p > 0.8$, $\bullet^1 p = 0.072$, $\bullet^2 p = 0.062$ $*^1 p = 0.034$, $*^2 p = 0.039$. Data are presented as mean values ± SEM. **d** Representative case showing the evoked field responses during basal conditions (blue), citralva (green), and conspecific male urine (orange), for PMCo (top) and hippocampal CA1 (bottom). Potentiation was only observed in the presence of urine, in both PMCo and CA1, but not during the exposure to the neutral odorant citralva. Each dot represents the average amplitude of six evoked potentials (±SEM). Calibration: vertical bar, 0.23 mV, horizontal bar, 120 ms. **e** Group data showing the lack of potentiation in PMCo (left) and CA1 (right) after the exposure to citralva (green), and the significant amplitude increase after the presentation of conspecific urine (orange); PMCo: $n = 6$ animals, CA1: $n = 7$ animals. Data represent the percentage of amplitude increase normalized to the average basal activity. Box plots are defined in terms of minima and maxima by whiskers, and the center and bounds of box by quartiles ($Q_1$–$Q_3$). Friedmann test with Dunn's post hoc comparisons, statistical tendency with ● $p < 0.08$, *$p < 0.05$, **$p < 0.01$. Exact $p$ values of Dunn's multiple comparisons are provided in Supplementary Table 1. Source data are provided as a Source data file.

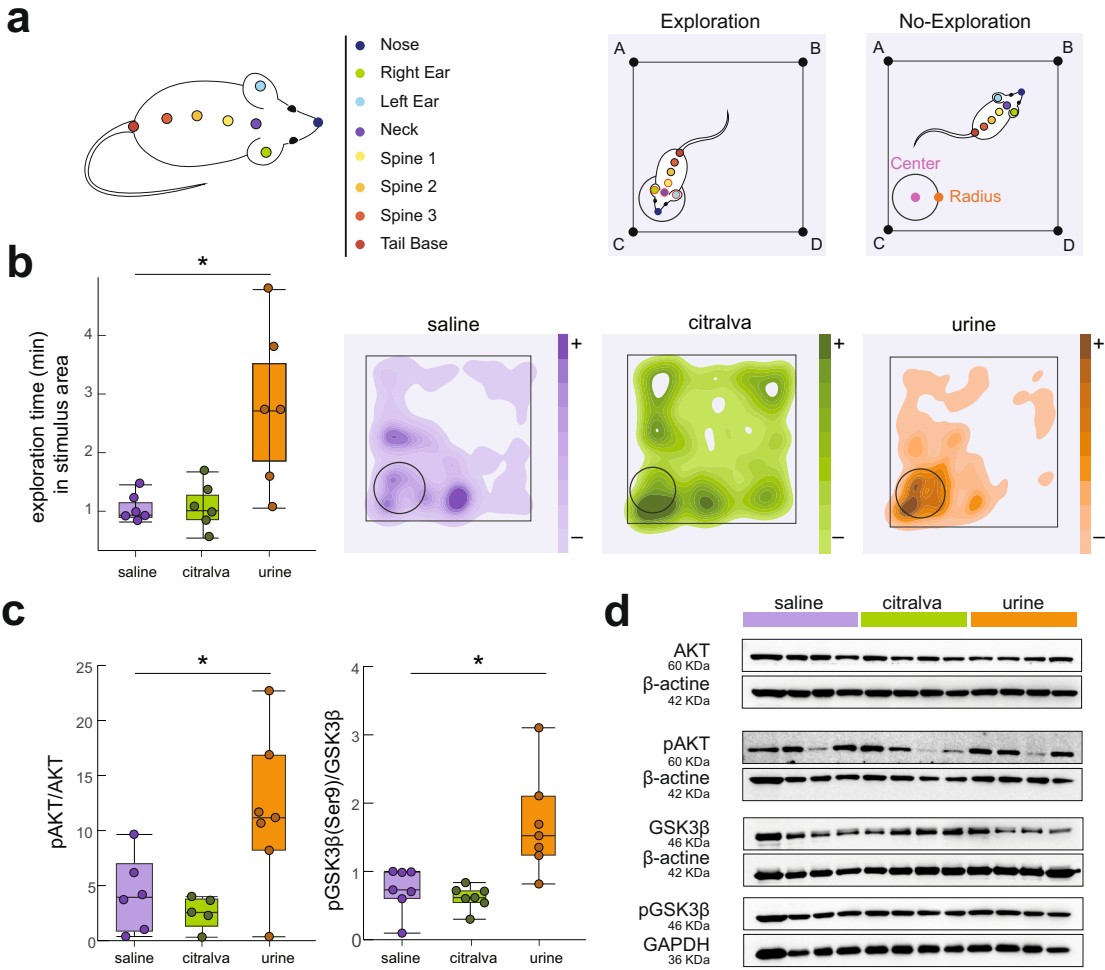

**Fig. 6 Urine of conspecific males induced a territorial preference and an increase expression of synaptic proteins in the hippocampus. a** Left: the labels used to train our DLC network both for the mouse and the arena. Right: exploration was quantified when nose, right ear, left ear, and neck were inside the region of interest. A train error of 2.09 pixels and test error of 2.91 pixels were achieved. **b** Exploration time (min) in the stimulus area in the first 10 min of the preference test. An ANOVA analysis with Tukey post hoc showed that urine stimuli induced significantly longer exploration time than citralva ($p = 0.0048$) and saline ($p = 0.0041$); $n = 18$ animals. Box plots are defined in terms of non-outliers' minima and maxima by whiskers, and the center and bounds of box by quartiles ($Q_1$–$Q_3$). Right: representative heatmaps of the animals in the saline (purple), citralva (green), and urine (orange) groups, showing the preference for exploring male urine. **c** Western blot measure of pAKT/AKT and pGSK3b/GSK3b in the dorsal hippocampus of the same animals used in the preference test. Urine presentation induced higher levels of pAKT/AKT than citralva ($p = 0.016$) or saline ($p = 0.037$), and a higher ratio of pGSK3b/GSK3β (urine–citralva: $p = 0.0015$; urine–saline: $p = 0.0038$); $n = 18$ animals. **d** Representative Western blot results: gel with 12 samples ($n = 4$ per group). An additional gel with six samples is shown in Supplementary Fig. 2a. For all panels: one-way ANOVA test with Tukey post hoc, *$p < 0.05$. Source data are provided as a Source data file.

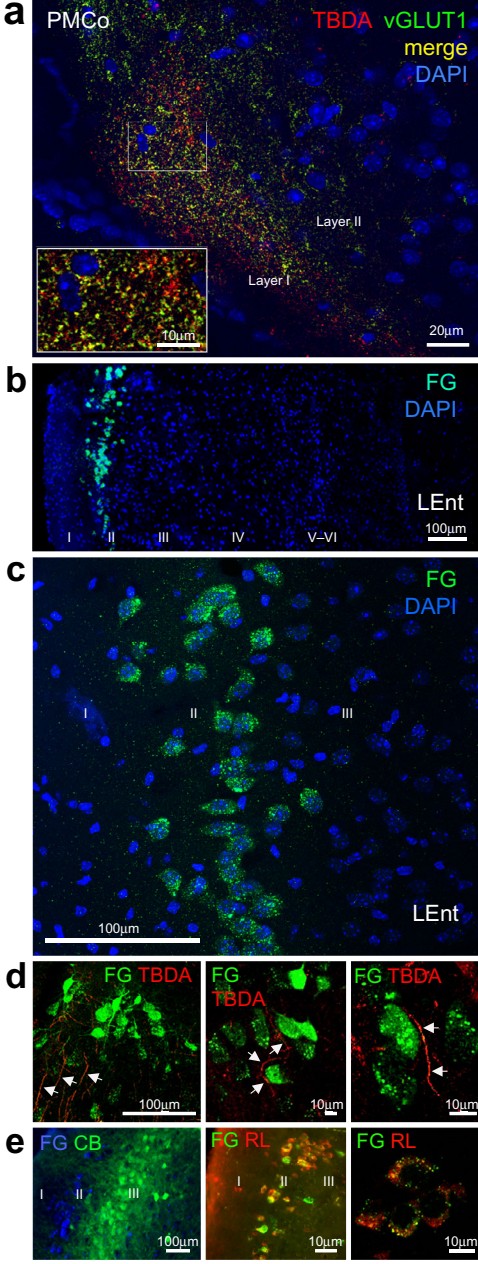

**Fig. 7 Anatomical evidence of the glutamatergic input from the accessory olfactory bulb (AOB) to the vomeronasal cortical amygdala (PMCo), and from the PMCo to the dorsal hippocampus. a** Confocal microscopy image obtained after anterograde tracer injections in the AOB (red) and immunofluorescence detection of the vesicular glutamate transporter 1 (green), showing numerous double-labeled terminal boutons (yellow) in the PMCo ($n = 4$ independent experiments). Inset from the squared area is shown. **b**, **c** Retrogradely labeled cells (green) in the layer II of the dorsal aspect of the lateral entorhinal cortex (dLEnt), following FluoroGold injections in the dorsal CA1 ($n = 8$ independent experiments). **d** Anterograde tracer injection in the PMCo gave rise to labeled fibers (red) around the retrogradely labeled cells (green) in the dLEnt that project to the dorsal CA1, evidencing the anatomical pathway from the amygdaloid PMCo to the hippocampal formation ($n = 6$ independent experiments). **e** Double-labeling of the retrograde tracer with calbindin (left) and reelin (middle and right panels) shows that only reelin-positive dLEnt neurons project to the dorsal CA1 ($n = 3$ independent experiments).

showed that male urine induced a significantly higher exploration time (Fig. 8a) compared to the neutral odorant ($t_8 = 3.59$; $p = 0.007$). The expression of c-fos was also significantly higher in the dLEnt (Fig. 8b; $t_8 = 4.34$; $p = 0.002$), in the area where the axonal projections of the PMCo overlap with neurons projecting to the hippocampal CA1. In addition, we investigated the proportion of double-labeled cells for c-fos and reelin in each group, and the results showed that it was also significantly higher in the experimental group (Fig. 8c; $t_8 = 3.51$; $p = 0.0001$). Furthermore, taking together all animals, the time spent exploring the stimuli showed a significant correlation with the number of double-labeled c-fos/reelin neurons (Pearson's $r = 0.71$; $p = 0.02$). As expected, male urine also induced a significantly higher c-fos expression in the PMCo (Fig. 8d; $t_8 = 2.48$; $p = 0.04$). In contrast, the number of c-fos-labeled cells in CA1 did not differ between the control and experimental groups (Fig. 8e; $t_8 = 0.26$; $p = 0.80$).

These results were confirmed reanalyzing preparations of a previous c-fos experiment, in which females were exposed to male-soiled bedding located in a particular area within the experimental cage[15], compared with females investigating clean bedding. Females exploring male pheromones displayed higher levels of c-fos than females in the control condition in the dLEnt ($t_{10} = 2.39$, $p = 0.038$), as well as in the PMCo (previously reported in ref. [15]). In this case, a significantly higher c-fos expression was also found in CA1 ($U = 4.5$, $p = 0.026$). In addition, a significant statistical correlation was found between the time spent investigating male-soiled bedding and the c-fos expression in PMCo and dLEnt (Supplementary Table 2). Thus, male chemical signals induced a significant c-fos expression in the PMCo and dLEnt independently of the use of a control group exposed to clean bedding or to a neutral odorant (citralva). In contrast, the CA1 showed higher expression only when the control group was exposed to clean bedding.

## Discussion

This study provides evidence that neuronal activity in the posteromedial cortical amygdala is coupled with that of the hippocampus, and the stimulation of the vomeronasal system results in synaptic plasticity in both areas. The hippocampal synaptic plasticity induced by the AOB–PMCo activation is probably triggered via the dLEnt, and therefore pheromonal information can influence hippocampal encoding.

The analysis of oscillatory activity in the vomeronasal amygdala and dorsal hippocampus during the exploration of the virtual corridors reveal that both structures show a high synchronic activity in the theta range. The causal analysis reveals a top–down control in which CA1 drives the population activity of PMCo, and also, to a lesser extent, time periods in which PMCo leads the CA1 oscillations. These results suggest that an intense transfer of information between the two areas takes place while mice navigate the virtual environment. Similar synchronic activity has been described in the basolateral amygdala–hippocampal circuit, where it plays a key role in the acquisition and extinction of emotional memories[16]. We described a similar coupling between the dorsal hippocampus and the vomeronasal cortical amygdala.

Although we expected that the synchronization of oscillatory activity between the PMCo and CA1 would increase in the presence of vomeronasal stimuli, our results show that coupled activity in the theta range is found along navigation through the different virtual corridors, independently of the presence of salient visual, olfactory, or vomeronasal stimuli. One possible interpretation is that animals are alert and actively exploring the virtual environment, and the vomeronasal amygdala is part of the circuit processing incoming information in this state. Consistent with this hypothesis, we have previously found that the main and

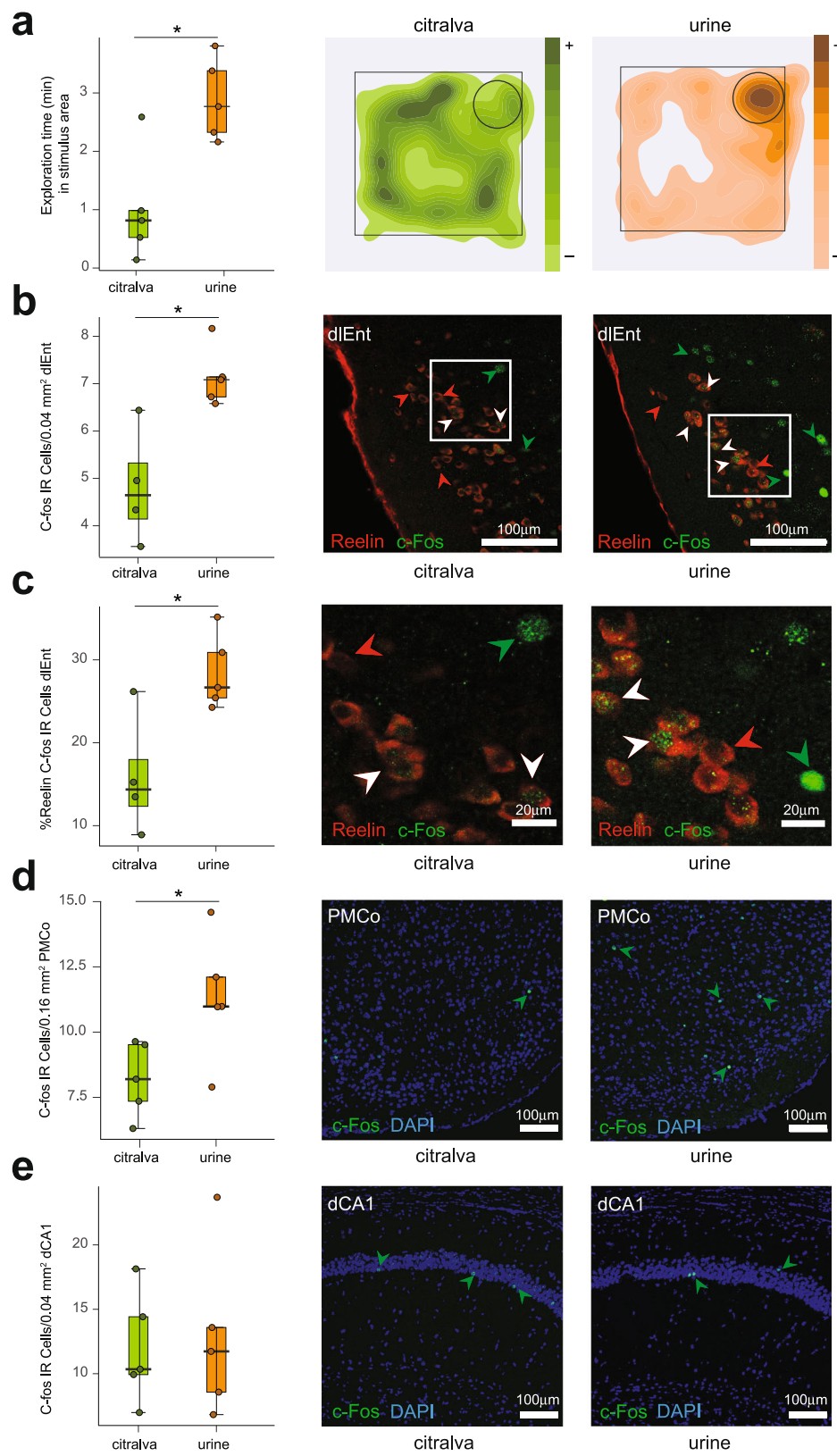

accessory olfactory bulbs show synchronic activity even when exploring neutral olfactory stimuli[17]. Thus, the vomeronasal system is probably actively engaged in the exploratory activity together with the olfactory and probably the visual and somato-sensory (through the vibrissae) systems. This suggests that exploratory behaviors involving multisensory cortical areas also

incorporate pheromonal information from the posteromedial amygdala.

We hypothesized that local gamma activity may show a more specific signature of processing vomeronasal information in the PMCo–CA1 network. Thus, we analyzed the gamma components nested in individual theta cycles[10]. This analysis reveals that a

**Fig. 8 Urine of conspecific males induced a territorial preference and an increased activity of dlEnt and PMCo cells. a** Exploration time (min) in the stimulus area in the first 10 min of the preference test. Urine induced significantly higher exploration time than citralva; $p = 0.007$; $n = 5$ animals per group. Box plots are defined in terms of minima and maxima by whiskers, and the center and bounds of box by quartiles ($Q_1$–$Q_3$). Right: representative heatmaps of the animals in the citralva (green) and urine (orange) groups, showing preference for male urine. **b** The number of c-Fos-IR cells/0.04 mm$^2$ and **c** % reelin c-Fos cells in dlEnt layer II was significantly higher in urine group (c-Fos, $p = 0.002$; reelin c-Fos, $p = 0.0001$; $n = 5$ animals per group). **b, c** Right panels: double immunofluorescence for reelin and c-Fos cells indicated with arrows; $n = 5$ animals per group. In **c**, a detailed view of the squared area in **b** can be observed. **d** In PMCo a significantly increased number of active cells was observed; $p = 0.04$, $n = 5$ animals per group. **d** Right: c-Fos-IR cells in the PMCo, indicated by arrows. **e** c-Fos-IR cells were not significantly more active in dCA1; $p = 0.80$, $n = 5$ animals per group. Right: c-Fos-IR cells in dCA1, indicated by arrows. Scale bar in **b**, **d**, and **e**, 100 μm; in **c** 20 μm. For c-Fos-IR cells, two-sided $t$ test; for % reelin c-Fos cells, beta regression analysis; *$p < 0.05$. Source data are provided as a Source data file.

particular pattern of theta-nested gamma components is present in the sector associated with urinary stimuli, which is common to PMCo and CA1, characterized by a fast-gamma rhythm (tSC5, >120 Hz). In contrast, salient visual stimuli elicited very different patterns of tSCs in the PMCo and CA1, which in turn are also different from that observed in the presence of urine.

Thus, these observations suggest that fast-gamma oscillations are strongly associated with amygdaloid processing of ongoing vomeronasal inputs. Accordingly, fast-gamma rhythms have been suggested to promote the transmission of current sensory information to the hippocampus during new memory encoding[18]. With experience, this fast-gamma gives rise to a high-gamma component (tSC4, around 80 Hz), especially in CA1, which could reflect an operating state of CA1 dominated by medial entorhinal inputs, probably related to the generation of spatial maps[10,19,20].

Synchronous oscillations in the basolateral amygdala–hippocampal circuit have been shown to facilitate synaptic plasticity related to threat learning[16,21]. Thus, the synchronic activity in PMCo and CA1 revealed by the LFP analysis might be indicative of synaptic plasticity in this circuit, allowing the integration of vomeronasal information in the hippocampal cognitive map. In fact, the present results demonstrate not only the induction of LTP in the vomeronasal pathway to the cortical amygdala, but also simultaneous synaptic plasticity in the dorsal hippocampus. LTP in the PMCo and CA1 can be induced using tetanic stimulation of the aot, and also with natural stimulation using male urine as stimulus. Similar to our results, LTP in the dentate gyrus has been induced with electrical stimulation of the anterior piriform cortex[22]. Although olfactory cues contained in urine are also probably playing a role, the LTP induction by urine in anosmic animals, both in the PMCo and CA1, shows that the vomeronasal signals are sufficient to induce this synaptic potentiation. Thus, the dorsal hippocampus is strongly influenced by the activity in the vomeronasal system, and pheromonal information contained in urine, probably chemical signals of conspecifics, induces synaptic plasticity in dorsal CA1, likely contributing in a relevant way to generate a territorial map which includes the identity of conspecifics. Vomeronasal chemical signals not only allow the recognition of conspecifics and their health status[23,24], but also the presence of predators[25,26]. This information is relevant for the animal to navigate the environment, and optimize the probabilities of survival and reproduction. Of note, vomeronasal information derived from either conspecifics or predators has an important emotional component, and from that perspective it is not surprising that it is processed in the amygdala and relayed to the hippocampus similar to what has been described for the amygdalo–hippocampal network in relation to threat learning. Therefore, the vomeronasal-induced plasticity in the hippocampus is probably a key feature for learning the spatial map of the environment and for its continuous updating.

Freely behaving female mice show a preferential exploration of male urine present in a particular location of the test cage, and develop place preference for this location[7,27]. We used this urine-induced preferential exploration to investigate the expression of markers of synaptic plasticity in the PMCo and the dorsal

hippocampus. The results show a significant increase of active Akt in the hippocampus of urine-exposed animals. This protein kinase is part of the signaling cascade of phosphoinositide 3-kinase (PI3K), which has been previously demonstrated to mediate synaptic plasticity, in the form of LTP, in the hippocampus[28,29]. LTP involves the activation of NMDA receptors and the PI3K–Akt pathway leading to decreased activity of GSK3β. GSK3β is a critical downstream component of the PI3K/Akt pathway, whose activity can be inhibited by Akt-mediated phosphorylation in Ser9, a crucial step in the induction of LTP[30]. In our findings we observed the increase of the inactive state of the GSK3β, consistent with the activation of the PI3K–Akt–GSK3β pathway, a key mechanism in the regulation of synaptic efficiency.

In contrast, our results did not reveal the involvement of pCREB/CRTC1 in the amygdala in urine-induced preference. The confluence of the two olfactory modalities on the amygdala could be masking a net effect of the influence of pheromones on these markers of synaptic plasticity. Alternatively, the time window for the expression of these synaptic markers in PMCo may differ from that in the hippocampus.

The results of our tract-tracing experiments confirm the lack of direct projections from the vomeronasal amygdala to the dorsal hippocampus, as previously reported in mice[8,14] and rats[12,13]. However, an indirect pathway links the PMCo with the dorsal CA1 through the dLEnt. A similar indirect pathway, from the anterior piriform cortex to the LEnt, has been suggested to mediate the influence of olfactory information into the hippocampus[22]. Thus, vomeronasal information would converge in the dLEnt with olfactory and other types of sensory information, thus reaching the hippocampal formation as a complex multisensory input. In fact, this projection has been shown to be involved in olfactory learning[31]. It should be noted that the region of the dorsal LEnt, where the neurons projecting to CA1 are located, receives a direct olfactory input from the main olfactory bulb[32], as well as an indirect olfactory input from the piriform cortex[33,34]. According to the results of the c-Fos experiment, reelin-positive neurons of dLEnt are specifically activated with urine. These neuronal population responds to olfactory stimuli and projects to the hippocampus[35]. Thus, these neurons are very likely integrating both olfactory and vomeronasal information.

To date, the influence of vomeronasal information in the hippocampal spatial map has been completely ignored in the abundant previous literature on the role of the rodent hippocampus in spatial learning[6]. This type of information is crucial to form a territorial map, which includes the recognition of conspecifics[4,5]. This map will need continuous updating during the life of the animal, and therefore synaptic plasticity in this circuit is required to learn changes in the territory owners and the alteration of territorial boundaries. This ability will be equally necessary for males, to be able to recognize their own territory and that of competitors, and for females, to learn the territories with better resources and the owner of each territory[2]. Therefore,

this circuit is the neural correlate of territorial behavior in mice, allowing the integration of pheromonal and spatial information.

The convergence of a complex multisensory stream to the hippocampus might allow the formation of context-dependent, conjunctive representations of "what" and "where" of the episodic memory[36]. The incoming of the pheromonal information in this functional pathway could mean the inclusion of the conspecific recognition into the memory, i.e., the "who" component of the episodic memory.

## Methods

Experiments were performed on adult CD1 mice ($n = 96$) with weights ranging from 30 to 60 g obtained from an official supplier (Animal Facility of the Central Unit of Research, University of Valencia, Valencia, Spain). Mice were kept on a 12 h light/dark cycle with constant ambient temperature ($22 \pm 1 \,°C$) and humidity, and water and food available ad libitum. Experiments were performed during the light phase of the cycle. The experimental procedures were approved by the Research Ethics and Animal Welfare Committee of the University of Valencia (A1538643423944, A1537787672795, A20201118161356, and A20201118201142) and are in accordance with the European Communities Council Directive (2010/63/UE) on the protection of animals used for scientific purposes.

**Headplate implantation surgery**. Mice ($n = 8$) were anesthetized with isoflurane, placed in a stereotaxic frame (Narishige) and maintained at 37–38 °C with a heating pad. Two screws (ground and reference) were fixed above the cerebellum in opposite hemispheres. Supported by four screws, a custom-made resin headplate was fitted to the skull. Trephine holes were drilled above the right dorsal hippocampus ($-2.2$ mm AP, 1.2 mm ML, and 1.4 mm DV from the brain surface) and right posteromedial cortical nucleus of the amygdala ($-2.80$ mm AP, 2.75 mm ML, and 5.5 mm DV), referenced from bregma, according to the atlas of Franklin and Paxinos[37]. Once the probes were in place, the craniotomy was covered with SILASTIC (Kwik-Sil, World Precision Instruments). The wires from the ground and reference screws were soldered to the pins of the electrode interface board. Mice were placed in a cage and monitored until fully recovered. Upon reaching the presurgical weight, mice were handled daily as a habituation protocol. After 3 days, the mice were fixed to a metal holder by means of the plate anchored to skull (Fig. 1a), which in turn was connected to a stereotaxic apparatus (Narishige).

**Implementation of the virtual environments**. The virtual environment was based in a virtual reality system based in a cylindrical treadmill for head-fixed mice, in combination with a curved visual display. The movements of the treadmill were captured by an infrared sensor and converted into movements of the images projected on the monitor, thus finally reproducing navigation in virtual environments (Fig. 1a). Additional 3D objects were placed on the walls of the corridors (extra-current signals; Fig. 1b). A square object (called as "curtain") was also inserted in the center of the maze to further separate the corridor into two compartments. The curtain was moved aside when the subject finished s2. At the end of s4 (Fig. 1b), a reward of 1% sucrose solution was delivered. After consumption the reward, a 1-min rest was allowed until the beginning of the next trial. Mice were trained to run progressively longer corridors. Maze position and mouse speed were monitored and synchronized with LFP recordings.

The order of testing corridors was always the same, aiming to present first the non-pheromonal olfactory stimulus (c2), then the urinary stimulus (c3) and finally only the context (c4).

**Electrophysiological methods and analysis**. The LFP of the dorsal hippocampus (CA1) and cortical amygdala (PMCo) were recorded with a teflon-coated steel monopolar macroelectrode (120 μm diameter; World Precision Instruments, Aston, Stevenage, UK), referenced against an indifferent electrode placed in the cerebellar epidural space. The final position was determined by maximizing the amplitude of the LFP in response to a mild tail pinch. Data were acquired using the 32-channel RHD2000 hardware (Intan Technologies) and the Open-Ephys system (http://open-ephys.org) with a sampling frequency of 30 kHz. Raw signals were then imported to the MATLAB development environment (The MathWorks, Natick, MA, USA) for off-line analyses, which were performed using built-in self-developed routines or standard MATLAB libraries as necessary.

Time series were firstly downsampled to 1000 Hz, $z$-scored and processed in the time-frequency domain by the continuous wavelet transform, as described by Torrence and Compo[38]. The power of the signal at each wavelet scale (frequency) was defined as the modulus of the wavelet coefficient. These values were normalized to avoid scale-dependent biased values[39].

For detecting synchrony in a precise frequency range between both signals we used the spectral coherence as the ratio between the wavelet cross-spectrum and the individual autospectrums of both signals. Its squared value varies from 0 to 1, meaning low and high linear frequential correlation. This first approach was complemented with causality measures. Specifically, Granger's causality methods, widely applied in neuroscience, analyze the flow of information between time

series[40]. This analysis enables us to observe the directed influence for both neural signals, using predictability and temporal precedence of the time series.

Spectral causal measure was calculated by the MVGC multivariate Granger causality toolbox (MATLAB[41]). On the causal spectra, maxima were identified to detect directionality between the signals and their frequency. To evaluate the differences in the pattern of causality between sectors and corridors, the proportion of events of each type of causality was calculated. As the proportions do not adjust to a normal distribution because they take continuous values between 0 and 1, we performed the transformation $\sin^{-1}\sqrt{p}$[42]. Then, we fit a generalized mixed effects linear model to assess statistical differences in the causality ratio along the virtual reality system[43], with Tuckey post hoc test when appropriate.

Derived from the previous time-frequency analysis under exploratory behavior, we specifically studied the tSCs. To this end, in this analysis, we follow the procedure proposed by Lopes-dos-Santos et al.[10]. Briefly, we first applied empirical mode decomposition (EMD), based in the Hilbert–Huang transform[44], where the signal is decomposed into intrinsic mode functions (IMFs) and a non-oscillatory residual. We delimited theta-band oscillations by combining the IMFs with mean instantaneous frequencies between 5 and 12 Hz. Moreover, low-frequency and supra-theta signals were defined as the sum of all the IMFs with mean frequencies <5 and >12 Hz, respectively. Theta cycles were selected by local maxima and minima of the theta time series from the EMD. A theta cycle was defined as a sinusoidal wave with a period between 71 ms (14 Hz) and no >200 ms (5 Hz), where the zero-phase corresponds to the first trough of the wave.

Next, we calculated the wavelet power spectrum of the supra-theta signal between 10 and 200 Hz for each theta cycle. The spectral signature of each cycle was defined as the mean amplitude of each spectrogram in the frequency domain. By principal component analysis, we extracted the first five components of the spectral signatures, which represented over 87% of the variance. The procedure continued with an independent component analysis (ICA) of the whole set of spectral signatures. To maximize computational power, we here used the FastICA algorithm from the scikit-learn package (http://scikit-learn.org/stable/), implemented in Python 3.2, and applied inside the MATLAB environment. The extracted independent components were defined as tSCs[10]. To classify a single-cycle spectral signature in a given tSC, the strength of this tSC was defined as the similarity between a single-cycle spectral signature and the representative tSC (inner product between a tSC and the spectral signature). A threshold for the distribution of the single-cycle tSC strengths was used.

With the estimation of the strength measures, we obtained a vector for each context (corridor/sector) and each channel (PMCo, CA1). Therefore, we achieved a whole matrix ($5 \times 32$) containing tSCs as columns and strength as rows, and with strength values normalized by rows. To compare the tSC profile for each context, we used an unsupervised hierarchical clustering capable of correlating similar profiles, arranged in a clustergram, and where similar profiles show near positions attending to the Euclidean distance (Fig. 3). Thus, we have a complete vision of the regions defined by the predominance of specific tSCs. A bootstrapping technique was used to assess the accuracy of the results, with probabilities calculated after 1000 iterations.

**Long-term potentiation methods**. For LTP experiments, animals were anesthetized with an intraperitoneal injection of urethane (0.7 g/kg) and a subcutaneous injection of lidocaine (0.2 ml; B. Braun, Germany) into the neck to facilitate tracheotomy and thus to reduce respiratory problems associated with anesthesia. Surgery was initiated after the loss of corneal–palpebral reflexes and response to painful stimulation, as well as a slow breathing rate. To ensure the fixation of the electrodes with dental polymeric cement, distilled water, acetone, and orthophosphoric acid were consecutively applied to the surface of the skull.

The anesthetized mice were secured to a stereotaxic frame (Narishige, Japan) and maintained at 37–38 °C with a heating pad. Following a midline sagittal incision, trephine holes were drilled guided by stereotaxic coordinates from Paxinos and Franklin[37] adapted to CD1 mice, according to the distance between bregma and lambda. The animals were implanted with two bipolar stimulation electrodes in the aot ($-0.9$ mm AP, 1.8 mm ML, and 5–5.5 mm DV) and the Schaffer collaterals ($-1.5$ mm AP, 2 mm ML, and 1.4 mm DV from the brain surface) in the dorsal hippocampus. Recording electrodes were located at the right posteromedial cortical nucleus of the amygdala ($-2.80$ mm AP, 2.75 mm ML, and 5.5 mm DV), the AOB (3.5 mm AP, 1 mm ML, and 1.6 mm DV), and the CA1 stratum radiatum ($-2.2$ mm AP, 1.2 mm ML, and 1.4 mm DV from the brain surface). The histology of the position of the electrodes is shown in the Supplementary Fig. 1a–d. The recording electrode in the AOB was only used to guide the final location of the stimulation electrode in aot, determined by the maximum response evoked in the AOB after the tract stimulation. To stimulate the aot without simultaneous stimulation of the olfactory fibers running in the lateral olfactory tract (lot), we reexamined the restricted injections of neural tracers in the main and AOB available in our laboratory (published in ref. [45]). The aot and lot run together until the level of the anterior amygdaloid area, where the aot fibers course medially toward the anterior medial amygdaloid nucleus and the bed nucleus of the aot. We chose the coordinates of this position, just anterior to the medial amygdala, where the aot fibers concentrate (Supplementary Fig. 1e) and very few olfactory fibers are present (Supplementary Fig. 1f).

The evoked potentials (fEPSP) were recorded with stainless steel polyimide-coated macroelectrodes, with an external diameter of 0.125 mm (E363/6/SPC; PlasticsOne, USA). For the stimulation, we used a 0.05 mm stainless steel bipolar electrode (E363/6/SPC; PlasticsOne, USA). Electrical activity was pre-amplified with Grass P55 amplifiers (Grass-Technologies, West Warwick, RI) with a bandwidth of 30 Hz–1 kHz, and amplified with a device Ampli4G21 (Cibertec S.A., Madrid, Spain) with a bandwidth of 300 Hz–10 kHz. The signal was digitized (Power 1401; Cambridge Electronic Design, UK) with a sampling frequency of 10 kHz, and monitored online using Signal software (Cambridge Electronics Design). For basal and poststimulus measures, each animal was presented with a single square pulse (100 µV) applied with a frequency of 0.1 Hz during 30 and 90 min, respectively. The stimulus intensity was set at 30% of the value necessary for evoking a maximum fEPSP response. For LTP induction, each animal was presented with HFS protocol consisting of two trains of six pulses (15 ms, 400 Hz) at a rate of 1 per 20 s, separated 5 min (males, $n = 12$ for PMCo, $n = 7$ for CA1).

In a different group of animals (females, $n = 11$ for PMCo, $n = 12$ for CA1), instead of tetanic stimulation of the aot, a drop of 30 µl of male urine was placed in the nostrils of the animal after recording the basal evoked potentials. During 30 min under urine stimulation, the evoked potentials were recorded in the PMCo and CA1, as described before.

We measured the amplitude of the evoked potential through the use of the "peak-to-peak" parameter defined in Signal software, and quantifying the difference between the maximum and the minimum peak of the evoked wave. These measures, obtained in the baseline period and after tetanic stimulation, were normalized by the average of the last 30 basal measures. For statistical analysis, the fEPSP values were grouped in sets of six neighboring measures and averaged, by comparing amplitude values obtained from the baseline period and after tetanic stimulation in terms of percent of responding. The amplitude values at 5–10, 10–15, and 20–25 min after eliciting stimulation were compared against the basal values by using the nonparametric Friedman's test to infer differences and Dunn's test for multiple comparisons. The same approach was used for male urine responses.

At the end of the recording and stimulation sessions, animals were deeply anesthetized with sodium pentobarbital (100 mg/kg; Dolethal Vetoquinol, Spain) and transcardially perfused with 80 ml of saline solution (0.9%) followed by 80 ml of paraformaldehyde (4%) diluted in PB (0.1 M, pH 7.6). Brains were removed from the skull, postfixed overnight at 4 °C, and stored in 30% sucrose in PB (0.1 M, pH 7.6) at 4 °C for cryoprotection. Coronal sections of 40 µm were obtained in a freezing microtome, mounted onto gelatinized slides, and Nissl-stained to check for electrode placement.

**Long-term potentiation in lesioned olfactory epithelium.** In order to lesion the olfactory epithelium, animals ($n = 11$ females) were first anesthetized with iso-fluorane. Once reflexes were lost, 8–10 µl of 10% ZnSO₄ were applied into each nostril using a pipette. Analgesics (buprenorphine) were injected intraperitoneally to prevent inflammatory pain.

To investigate the presence/absence of olfactory recognition after the olfactory epithelium lesion, the habituation–dishabituation test[46] was carried out in lesioned ($n = 10$) and control ($n = 9$) animals. This test was performed inside a methacrylate box ($23 \times 23 \times 30$ cm) with a hole at a height of 8 cm through which a cotton swab impregnated with 20 µl of water, citralva (8,7-dimethyl–2,6-octadiene-1-nitrile, International Flavors and Fragrances, Barcelona, Spain), or urine (CD1 urine preserved for 48 h at 4 °C) was introduced. Citralva was used as a control odorant stimulus because it is a synthetic odor not involved in intraspecific communication[27]. Animals were placed for 5 min in the arena to habituate. Afterward, they were exposed to six consecutive 1-min presentations of each stimulus, following a sequential order.

At the end of the LTP experiments (PMCo, $n = 6$; CA1, $n = 7$, some animals were used for LTP in only one of the structures) 5 µl of DiI (Vybrant DiI, ThermoFisher) were pipetted into the ipsilateral nostril of the urethane-anesthetized mouse to assess whether it may be incorporated by the VNO. After transcardial perfusion, mice snouts were postfixed with 4% PFA in 0.1 M PB (pH 7.6) for 48 h. Then, VNO was carefully dissected, washed with PBS, and decalcified using 250 mM EDTA in 0.1 M PB during 5 days at 4 °C. Once the surrounding bone was soft, VNOs were embedded in 15% gelatin in 0.1 M PB at 4 °C overnight, trimmed, and incubated in 4% PFA in 0.1 M PB for 2 h at 4% PFA. Gelatin blocks were then cryoprotected with 30% sucrose in 0.1 M PB at 4 °C and 30-µm-thick coronal sections were obtained in freezing microtome. Finally, slices were counterstained with DAPI (1 µg/ml, Molecular BioProducts) in distilled water for 5 min and mounted with FluorSave™ (Merck Millipore). Fluorescent images of VNO sections were then obtained with Leitz DMRB microscope epifluorescence (LEICA EL-6000) equipped with specific filters for DAPI (LEICA, A) and DiI (Leica N2.1), and were imaged using a digital LEICA DFC495 camera.

**Molecular analysis.** To investigate the molecular correlate of the induction of synaptic plasticity in the PMCo and CA1 by urine stimuli, we tested the preference of freely behaving CD1 female mice to investigate male urine ($n = 6$), citralva ($n = 6$), or saline ($n = 6$) during a 90-min test. Preference tests were performed in individual cages ($220 \times 220 \times 145$ h mm, Panlab), where animals were previously housed. Stimuli were presented in glass dishes (6 cm diameter and 5.5 cm high),

placed in the bottom-left corner, containing 5 g of clean bedding impregnated with 30 µl of adult male urine (CD1 urine preserved for 48 h at 4 °C), 0.5 µl of citralva, or 30 µl of saline. Three days prior to the test, animals were habituated during a daily 10-min session. During habituation sessions, volatile stimuli were introduced in the environment by placing, respectively, 30 g of male-soiled bedding; 30 g of bedding impregnated with 100 µl of citralva; or 30 g of bedding impregnated with 100 µl of saline, ~30 cm outside of the animal cages. These stimuli were not visible, as cages were surrounded by black walls to avoid visual cues. All behavioral experimentation was performed within a sound-attenuated room at stable temperature (23 °C) and light conditions (≈60 lux).

At the end of the test, animals were sacrificed by cervical dislocation. The dorsal hippocampus and the region encompassing the posteromedial cortical amygdala were bilaterally dissected and processed for western blot analysis. Tissue was homogenized with lysis buffer: 76.5 mM Tris (pH 6.8); 2% SDS; 10% glycerol supplemented with 2 mM sodium orthovanadate and protease inhibitor (Sigma-Aldrich) in a proportion of 1 ml lysis buffer/100 mg brain tissue; and using a mechanical shear with a Potter-glass-teflon homogenizer (Rw20 DZM Homogenizer, Janke & Kunkel), at 2000 r.p.m. in ice. Then, the concentration of proteins was determined by Lowry method measuring the absorbance of the samples and extrapolating them to the precalculated BSA pattern. Samples were incubated in loading buffer 2× overnight at 4 °C. A total of 20 µg of protein was loaded into each well in 12.5% acrylamide gels and electrophoresis was performed for 60 min at 100 mV. Subsequently, samples were transferred into nitrocellulose membranes for 90 min at 240 mA. The nitrocellulose membranes were blocked with 5% BSA (w/v) in 1× TBS-Tween (TBS-t) for 1 h at room temperature, and were incubated overnight at 4 °C with primary antibodies in the correspondent blocking solution: Akt (9272, Cell Signaling Technology, 1:1000); pAkt (9271, Cell Signaling Technology, 1:1000); CRTC1 (2587, Cell Signaling Technology, 1:1000); pCREB (9191, Cell Signaling Technology, 1:500); GSK3β (JM-3494-100, MBL, 1:100); pGSK3β(Ser9) (05-643, Merk, 1:50); and β-actin (A1978, Sigma-Aldrich, 1:500) and GAPDH (G9545, Sigma-Aldrich, 1:20000), which were used as the loading control. The next day, membranes were washed three times with TBS-t for 20 min in movement, and incubated in secondary antibody for 1 h: anti-mouse IgG H&L Chain Specific Peroxidase Conjugate (401215, Calbiochem, 1:6000); or anti-rabbit IgG HRP-linked (7074 S, Cell Signaling, 1:3000). Signal detection was performed using "Luminata Classico Western HRP Substrate" (WBLUC0500, Millipore Corporation, Billerica, USA). Western blot images were developed with a biomolecular imager (ImageQuant™ LAS 4000, GE Healthcare Bio-Science). Densitometry of western blot images was accomplished using ImageGauge4.0.

**Histological analysis.** To assess the effect of urine stimuli on the activity of the dlEnt, PMCo, and CA1, we tested the preference of freely behaving CD1 female mice to investigate male urine ($n = 5$); or citralva ($n = 5$) during a 90-min test, following the same behavioral paradigm described for the molecular analysis.

At the end of the test, animals were transcardially perfused and tissue was processed as previously described. Selected slices were treated with 1% sodium borohydride in 0.05 TBS (pH 7.6), and then blocked with 4% NGS (G9023, Sigma-Aldrich) and 0.3% Tx-100 (PanReac AppliChem) in TBS. After that, sections were incubated in primary antibodies with 2% NGS and 0.3% Tx-100 in TBS overnight at 4 °C: mouse anti-Reelin (MAB5364, EMD Millipore, 1:1000); rabbit anti-c-fos (SC-52, Santa Cruz Bio Tech, 1:500). Slices were then incubated in corresponding secondary antibodies for 2 h in blocking buffer: Rhodamine RedTM-X goat anti-mouse IgG (R6393, Life Technologies, 1:200) and Alexa Fluor 488 goat anti-rabbit (111-545-003, Jackson ImmunoResearch, 1:200). Finally, sections were counterstained with DAPI (1 µg/ml, Molecular BioProducts) in distilled water for 5 min.

Fluorescent sections were mounted with FluorSave™ (Merck Millipore) and photographed in a Fluoview FV1000 confocal microscope (Olympus). Three-channel (405, 488, and 559 nm laser wavelengths) image files were obtained maintaining unique excitation and acquisition parameters for all samples. Quantitative data was obtained by automated digital image analysis with FIJI software. For c-fos expression level assessment, a binarized 16-bit image of the channel 488 was obtained by applying a 0.7% threshold of the highest intensity value for each image. Thresholded particles above 10 pixels were measured and numbered through "outlines" FIJI function, thus generating a new image of c-fos positive particles. ROI was delineated using DAPI channel as reference. This image was overlapped to DAPI channel and c-fos inmunoreactive neurons were then counted as positive when three or more c-fos positive particles match in the same DAPI positive nucleus. Results were expressed as normalized area fraction. The same workflow was followed for dlEnt layer II reelin (channel 559) determination by applying a 3% threshold of the intensity and automatically counting particles above 150 pixels. Previous c-Fos-IR cells were then compared to reelin IR cells obtaining a total of reelin activated cells. Results were expressed as a percentage of reelin activated cells relative to total reelin IR-neurons analyzed. For comparisons of c-fos/reelin proportions, a beta regression model was used[47].

**Behavioral analysis.** Exploratory behavior was quantified with DeepLabCut[48]. For the habituation–dishabituation test, we created a network to track points of interest described in Fig. [6] from recorded videos (30 f.p.s.). Twenty frames were extracted from 12 randomly selected videos using $k$-means clustering to represent behavioral

diversity observed. The network was trained for 240,000–340,000 iterations within the computer cluster of the Bioinformatics and Biostatistics Unit in Principe Felipe Research Center (CIPF, Valencia, Spain). From each of the training videos, we extracted 20 outlier frames, with points with a $p < 0.9$ being relabeled. The network was then refined using same number of iterations, achieving a final train error of 3.29 pixels and a test error of 5.37 pixels. For the preference test, we created a network to track same points of interest described in Fig. 7. This network was trained for 400,000 iterations, and refined as described before, achieving a final train error of 2.09 pixels and a test error of 2.91 pixels.

Subsequent data generated by DeepLabCut was then processed using self-written Python code. For the habituation–dishabituation test, exploration was considered positive when the distance between animal's nose and the end of the cotton swab was <0.25 time the total distance. For the preference test, the area of the glass dish was defined as the ROI. Exploration was considered positive when "nose", "right ear", "left ear", "neck", and "spine 1" were all within the ROI simultaneously.

**Tract-tracing methods**. For tract-tracing experiments, animals (seven males and four females) were anesthetized with isoflurane (2.5–3%) in oxygen (0.8–1.5 l/min; MSS Isoflurane Vaporizer, Medical Supplies and Services, UK) delivered through a mouse anesthetic mask attached to the stereotaxic apparatus (David Kopf, 963-A, Tujunga, CA, USA). Analgesia was provided by butorphanol (5 mg/kg; Torbugesic vet, Zoetis S.L., Madrid, Spain), and atropine (0.05 mg/kg in distilled water; Atropine Sulfate Salt, Sigma-Aldrich, MO, USA) was used to reduce respiratory problems. During surgery, mice rested on a thermal blanket and eye drops (Siccafluid, Thea S.A. Laboratories, Barcelona, Spain) were used to prevent eye ulceration.

To study the connections between the vomeronasal and hippocampal pathways, mice received two different tracer injections in the right hemisphere. Tracer injections were performed with glass micropipettes with an inner diameter tip of 20–30 μm. First, an iontophoretic injection of the fluorescent retrograde tracer FG (hydroxystilbamidine bis (methanesulfonate), 4% in distilled water; Sigma-Aldrich, Cat #39286, MO, USA) was placed in the CA1 stratum radiatum of the dorsal hippocampus. Then, a second iontophoretic injection of the anterograde tracer TBDA (10000 MW, lysine fixable, 5% in PB 0.01 M, pH 7.6; Invitrogen, CA, USA) was placed in the PMCo (for CA1 and PMCo coordinates see above). To perform these iontophoretic injections, positive pulses (7 s on/off; 5 μA) were applied over 8 min using a current generator (Stoelting Co., IL, USA). To avoid diffusion of the tracer during the entrance and the withdrawal of the micropipette, a continuous negative retaining current (−0.5 μA) was applied, and the micropipette was left in place for 2 min after finishing the injection. After the injections, we closed the wound with Histoacryl (Braun, Tuttlinger, Germany). Two of the animals received bilateral injections of TBDA in the AOBs and were used for double-labeling with the glutamatergic marker vGLUT1 (Cat #135302, Synaptic Systems, diluted 1:1000).

After 7 days of survival, we anesthetized, perfused the animals, and processed the brains as described above for the LTP studies, and five parallel series of 40-μm-thick coronal sections were obtained with a freezing microtome.

Since FG could not be directly visualized in our confocal microscope, it was revealed using a rabbit anti-FG antibody (Millipore, Cat #AB153-I, RRID: AB_90738, Temecula, CA, USA) diluted 1:1000 in TBS-Tx with 2% NGS overnight at 4 °C and an Alexa 488-conjugated secondary goat anti-rabbit (Jackson Inmunoreserach, RRID: AB_2338046). TBDA was red fluorescence, and DAPI was used in some of the sections to visualize cytoarchitecture. Then, sections were mounted onto gelatinized glass slides, and cover-slipped with fluorescence mounting medium (Dako, Santa Clara, CA, USA). Additional series were processed for simultaneous visualization of FG and reelin, or FG and calbindin, using the same immunofluorescence protocol previously described (anti-reelin antibody: Cat #MAB5364, Millipore, dilution 1:1000. Anti-calbindin antibody: Cat #CB38, SWANT, dilution 1:1000). Confocal images were obtained using a Leica TCS SP8 microscope. Illustrations were designed with Adobe Illustrator (Adobe Systems, MountainView, CA, USA).

To evaluate the expression of c-fos induced by male chemical signals in dorsal CA1 and LEnt, we analyzed the slides of the immunohistochemical detection of c-fos in a group of female mice exposed to male-soiled bedding ($n = 6$) or clean bedding ($n = 6$), available at our laboratory (results of the amygdala published in ref. [15]). The level of dorsal hippocampus chosen to perform the analysis of Fos-immunoreactive (Fos-IR) neurons was the same where we placed the recording electrode in the LTP studies, and the level of the dorsal LEnt was the same where the tract-tracing studies indicated the location of the neurons retrogradely labeled following CA1 FG injections. Briefly, CA1 and LEnt pictures were captured with a 10× objective using a digital Olympus XC50 camera attached to an Olympus CX41RF-5 microscope. Photographs were taken maintaining constant light intensity and were processed with the Image J software (v.1.50i, NIH, MD, USA). A protocol for automated counting of stained nuclei (similar to that used in ref. [15]) was applied to all the samples. Briefly, images were binarized using as threshold the 85% of the gray level mode. Then, a ROI consisting of a rectangular counting frame of 0.1 mm² was selected. Elements inside the ROI were automatically counted if they were integrated by 35 or more pixels. The number of Fos-IR cells in the CA1 and LEnt was measured in both cerebral hemispheres and the density of Fos-IR

cells obtained from both measures was averaged and used for subsequent statistical analyses.

To confirm the connection between the dlEnt and the PMCo, iontophoretic injections of the fluorescent retrograde tracer FG (hydroxystilbamidine bis (methanesulfonate), 4% in distilled water; Sigma-Aldrich, Cat #39286, MO, USA) were performed in the dlEnt at levels where anterograde labeling from the PMCo was previously observed. After 7 days of survival, animals transcardially perfused and tissue was processed as previously described.

**Statistical analysis**. Statistical analysis was performed in Rstudio (1.4.1106). Applicability conditions for parametric tests were assessed with Shapiro–Wilks normality test tests (R Core Team, 2009) and Levene's test for homogeneity of variances.

**Reporting summary**. Further information on research design is available in the Nature Research Reporting Summary linked to this article.

## Data availability
The raw data of the LFP recordings of the experiments in the virtual reality setup generated in this study are available in the Zenodo database under accession code https://doi.org/10.5281/zenodo.5153116. The data generated in the LFP experiments (causality measures and theta components), long-term potentiation experiments, behavioral results, protein quantification using western blots, and c-Fos quantification are provided in the Source data file. Source data are provided with this paper.

## Code availability
All code used in the analysis is available from the corresponding author on reasonable request.

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

## Acknowledgements

Funded by the Spanish Ministry of Science and Innovation-FEDER (BFU2016-77691-C2-2-P and PID2019-108562GB-I00). We thank Liset Menendez de la Prida and Manuel Valero for the original design of the virtual reality system, and CIBERTEC SA for its implementation. We also thank Fernando Martínez-García and Jose Moncho-Bogani for the c-fos preparations, and Vitor Lopes-dos-Santos for his help with the method of analysis of the theta-nested gamma components; Mario Sendra and Andrés Bravo Nuñez for helping with statistical analysis; and Julia Corell Sierra for supervising Python scripts developed for DLC analysis.

## Author contributions

M.V.-F., E.L. and V.T.-M. designed the study. M.V.-F., M.E.V.-M., E.M., A.T.-S. and D.E. performed the experiments. M.V.-F., M.E.V.-M., D.E., E.M., A.T.-S., A.L., J.M.-R., A.C.-F., E.L. and V.T.-M. analyzed and interpreted data. M.V.-F., M.E.V.-M., V.T.-M. and E.L. prepared the manuscript draft. All authors revised, modified, and approved the final manuscript. E.L. and V.T.-M. supervised the study and obtained funding.

## Competing interests

The authors declare no competing interests.
