## [Peer Review File · Nature Communications]

Integrating pheromonal and spatial information in the amygdalo-hippocampal networkREVIEWER COMMENTS

Reviewer #1 (Remarks to the Author):

The authors address the hypothesis that pheromone signals are incorporated in/modulate the hippocampal map by investigating the potential functional and neuroanatomical interactions between the vomeronasal system and the dorsal CA1 field of the hippocampus. This study includes a coherent set of experiments ranging from behavior (virtual environment task, olfactory preference) to neurobiological measures (molecular substrate of LTP) that dissects and provides original findings on the vomeronasal/hippocampus interactions. This study certainly fills an important gap in the knowledge of these relationships by showing effects induced by vomeronasal stimulation in both PMCo and CA1. In particular the findings include gamma/theta oscillations cycle-by-cycle interaction in PMCo, urine-induced long term potentiation in PMCo, molecular mechanisms underlying LTP in PMCo, identification of the pathway from PMCo to CA1. To my opinion, the studies are well conducted and analyzed with relevant statistical analyses. Below are more specific comments.

I think that the current title reflects more an interpretation of the results than the actual results: Indeed, it is not clear how the paper demonstrates integration of pheromonal and spatial information. The spatial context (visual cues) is not manipulated and I am not sure to what extent this context is important for the task. I think that the use of a spatial representation is not critical for performing the task. Would removing the visual cues (or changing these cues) affect the LFPs coupling in PMCo/CA1?

Rats were trained in a virtual reality behavioral task during which they were exposed to a pheromonal stimulus or neutral olfactory stimulus. LFP were simultaneously recorded in the PMCo and CA1 during the task. Both regions displayed theta oscillation and it is mentioned in the text that 35% of the time the two regions are coupled.

- How is this coupling related to the task (more concentrated during S3 ? Even in C4S3 ?).

They used a method from Lopes-dos-Santos et al. (which was applied also to recordings in mice CA1) to perform single-cycle analysis of gamma/theta interaction in both the PMCo and the CA1 hippocampus. This analysis is very interesting and allowed to extract several spectral components in individual theta cycles in the present study, therefore contributing to validate the Lopes-Dos-Santos et al.'s analysis. In addition it shows that there are theta-nested gamma components in Amygdala and that a tSC5 fast gamma component in both Amygdala and hippocampus which is maintained across days 1 and 2.

- I wonder to what extent is the tSC5 specific to vomeronasal stimulation as it was also predominant when a neutral stimulus was applied? (C2 in day 2).

- It is argued that the shift from tSC5 to tSC4 during C4S3 reflects the spatial component. Could this shift be simply due to an absence of vomeronasal/olfactory stimulation?

- It appears intriguing that the tSC5 to tSC4 shift was seen in the hippocampus but not in the PMCo which is expected to be modulated by changes in vomeronasal inputs.

I have no particular comment for the LTP experiments that convincingly show LTP in the PMCo, a novel result, and indicate that vomeronasal stimulation by male urine induced LTP in PMCo and CA1.

P16. Methods section, the "final location of the recording electrode..." should be "stimulation electrode" ?

Reviewer #2 (Remarks to the Author):

This is an interesting paper that makes an original contribution to the literature in both the fields of hippocampal learning and chemosensation. These findings suggest the importance of a pathway from vomeronasal system to hippocampal associative learning circuits, via the posteromedial cortical amygdala and entorhinal cortex. They speculate that activity in this pathway underlies the relational learning of spatial and identity information that underlies mouse territorial behaviour. A strong point of the study is the use of natural chemosignals present in urine, which have previously been shown to act as unconditioned stimuli for chemosensory learning. The fact that at least some of the mice appeared to show long term potentiation of synaptic transmission in the posteromedial cortical amygdala and in CA1 of the hippocampus is a novel and highly interesting

finding. It is consistent with the changes in molecular markers of plasticity.

However, there does appear to be a weakness in the way in which the LTP data have been analysed. Thirteen male mice were used for electrically induced LTP and 15 females for urine induced LTP. But there are only 6 to 8 datapoints in each of the figures. Individual animals contribute to the data points in each figure, but it is not clear whether the data points in different figures relate to the same animals. The authors state in the methods "Each case was classified as LTP if the amplitude value, obtained in the 20-25 min period after tetanic stimulation, was above the first quartile in the boxplot distribution. For each one of the selected cases... were compared against the basal values...". Unless I'm mistaken, this implies that the authors have selected cases that show LTP and then analysed whether the selected cases show LTP in comparison with baseline values and in comparison with the 100% amplitude level. This would invalidate the statistical analysis, as the data have been selected and not randomly sampled. If this is the case it would render these statistics meaningless. Instead, I think that the authors need to be clearer on the proportion of the animals that showed LTP in each condition and maybe show results from all animals.

I've spent some time trying to understand the utility of the clustering analysis shown in figure 1. But I must admit that I find it very confusing and it doesn't contain any indication of error. Wouldn't it be simpler to replace it or supplement it with strength changes in the tSCs just for sector 3 of the different runways, along with an estimate of error or confidence?

Other points to consider include:

In the abstract the authors claim to find "...synchronic activity in the vomeronasal amygdala and the dorsal CA1 of the hippocampus" But they need to be more specific about what was synchronised, as it would not be clear to a reader what this "synchronic activity" means.

Line 109 "...this synaptic plasticity is relayed to the dorsal hippocampus,.." I don't think that you can say that synaptic plasticity is relayed between these brain areas. The evidence suggests that exposure to urine induces synaptic plasticity in both regions, but doesn't imply transfer of plasticity.

Line 275, Figure 2 legend, the authors state "induced by the tetanic stimulation of the accessory olfactory tract (A, B) and urinary pheromones (C, D)" but it would be better to refer to this as urinary stimulus (C, D), rather than pheromones, as pheromones not used specifically. The time axis for the boxplots is also unintelligible in figure 2.

Line 120 "and lower levels of activated GSK3 β in CA1 in urine-exposed animals than in controls (Fig. 3B; Extended Data Table 2)." The authors could make it clearer earlier on in the text that ratio of phosphorylated to non-phosphorylated GSK3 β is increased, which implies a decrease in active GSK3 β .

In Fig 3, it looks as though the authors have made a composite image of the gels with bands cut out of the original gel image. Would it not be better to show the original image for the whole gel?

In Fig 4a, it would be easier to assess co-localisation if there was a higher magnification insert in the figure.

The methods should really state which phase of the light dark cycle experiments were performed. I also don't find it clear what happens in the training corridor, is there just an absence of any stimulus so the animals are just trained to run?

Were the order of testing corridors counterbalanced across individuals or always run in the same order?

Line 418, In the methods, the authors state that recording electrodes implanted in the AOB but then there is no mention of why this was done or what was found.

Line 420, "The final location of the recording electrode in aot was determined guided by the maximum response evoked in the accessory olfactory bulb" I think this should be stimulating electrode, as there was no recording electrode in the aot.

Line 452, "For statistical analysis, the fEPSP values were grouped in sets of 5 measures and averaged,..." It's not clear what's meant by "grouped in sets of 5 measures". Does this mean 5 neighbouring timepoints were averaged?

Line 464, The authors state that they collected histological sections from the LTP animals. But they don't say any more about why they were collected or how they were used. Were they analysed for electrode placement? Could this have been used to select animals to analyse rather than the presence of LTP?

Peter Brennan

Reviewer #3 (Remarks to the Author):

The manuscript from Villafranca-Faus et al. reports a new interesting view that correlates chemosensory-induced activity in the cortical amygdala and parallel activation of the dorsal CA1 of the hippocampus, likely related to spatial learning. Although experiments are well conducted, conclusions are often not fully supported by provided data, in particular claims of pheromone influence on hippocampal learning. I find the work lacks some critical controls.

Major issues:

1. The link with the vomeronasal system is based mostly on circumstantial evidence. The PMCo largely receive vomeronasal inputs, but it cannot be excluded that other sensory information (i.e. olfactory) participate in modulating hippocampal activity, even from the same location. The same for the LEnt. Use of a neutral olfactory stimulus (sesame oil) in Fig. 1 goes in the right direction, but is not robust enough. To better assess pheromone and vomeronasal roles, it would be important to include a control with a mouse model deficient for vomeronasal function (genetic or surgical VNO ablation). If the authors' hypothesis is correct, a VNO-deficient mouse would retain full response after tetanic stimulation in the paradigm of Fig 2, but responses after urine stimulation will be lost in both the aot and hippocampus. Some level of activity still present at any of these two levels would be indicative of partial involvement of the main olfactory system and not only the vomeronasal system. Shown activity after application of urine rather suggest a role of the main olfactory system as VNO pumping is likely not active in anesthetized animals. Using purified pheromones instead of a complex mixture like urine may also elucidate this point.
2. In Fig. 3, the expression analysis of proteins related to synaptic plasticity in the amygdala and dorsal hippocampus is not clear to me and needs further validation. First, using an olfactory neutral (non-pheromonal) stimulus to demonstrate that olfactory stimulation alone is not sufficient to increase expression. Confirmation of the results using a second alternative method (qRT-PCR, etc.) would considerably support the conclusions.
3. Fig. 4 lacks measurements or quantification data. Do the authors find other overlapping brain areas (apart from dLEnt) of retrogradely labeled cells and anterogradely labeled fibers? At what extent? What proportion of cells from the dLEnt receives innervation from the PMCo? How many of those send projections to the hippocampus?
4. c-Fos expression in the dLEnt in females exposed to male pheromones is an important result. It would be helpful to show representative images as well as raw measurements of stimulated vs controls, and not only a correlation chart. Are these cFos+ cells in the dLEnt also positive for reelin?

Minor:

A scheme of the experimental preparation of Fig. 1 with the head-fixed mice, the different virtual navigation corridors and sequential stimuli will help to understand an otherwise complex figure.

POINT BY POINT REPLY TO REVIEWERS

Reviewer #1 (Remarks to the Author):

The authors address the hypothesis that pheromone signals are incorporated in/modulate the hippocampal map by investigating the potential functional and neuroanatomical interactions between the vomeronasal system and the dorsal CA1 field of the hippocampus. This study includes a coherent set of experiments ranging from behavior (virtual environment task, olfactory preference) to neurobiological measures (molecular substrate of LTP) that dissects and provides original findings on the vomeronasal/hippocampus interactions. This study certainly fills an important gap in the knowledge of these relationships by showing effects induced by vomeronasal stimulation in both PMCo and CA1. In particular the findings include gamma/theta oscillations cycle-by-cycle interaction in PMCo, urine-induced long term potentiation in PMCo, molecular mechanisms underlying LTP in PMCo, identification of the pathway from PMCo to CA1. To my opinion, the studies are well conducted and analyzed with relevant statistical analyses.

Authors' response: We thank the reviewer for these comments

Below are more specific comments.

I think that the current title reflects more an interpretation of the results than the actual results: Indeed, it is not clear how the paper demonstrates integration of pheromonal and spatial information. The spatial context (visual cues) is not manipulated and I am not sure to what extent this context is important for the task. I think that the use of a spatial representation is not critical for performing the task. Would removing the visual cues (or changing these cues) affect the LFPs coupling in PMCo/CA1?

Authors' response: To clarify how the work demonstrate integration of pheromonal and spatial information, we now explained in more detail that different visual cues characterize each one of the four sectors in the virtual reality task. These cues allowed the experimental animals to represent the different spatial contexts, which in turn may be associated with the presence of new visual cues (sector 2) or chemosensory stimuli (sector 3). We have added a new Figure 1, showing the configuration of the four corridors, and a supplementary video (Supplementary movie 1) of the virtual navigation. The presence of visual cues in sector 2 does not change the LFPs coupling in PMCo/CA1, but it changes the causality (Granger analysis, see next response) and it also changes the pattern of theta-nested gamma components (tSCs). In this regard, the analysis of the tSCs in each one of the virtual sectors shows that the presence of new visual stimuli, in corridor 2-sector 2, elicited a different pattern of activity in PMCo and CA1 (the two structures did not cluster together in any of the testing days). In contrast, the presence of urinary stimuli in corridor 3-sector 3 induced a shared pattern of tSCs in PMCo and CA1, characterized by a strong tSC5 and a relatively low activity in the rest of the gamma frequencies. Notably, the visually-elicited activity patterns of activity in the PMCo and CA1 are very different from the shared pattern observed at PMCo

and CA1 when vomeronasal stimuli are present. These activity patterns are now described in detail in the Results. In addition, in the new version of Figure 3 (old Figure 1), we have incorporated a detailed view of the pattern of tSCs in the PMCo and CA1, including the Euclidean distances between these patterns calculated for the clustering analysis. As suggested by reviewer #2 (see below), we have added an index of confidence of the obtained clustering (cophenetic correlation), and a bootstrap analysis to further validate the results. Thus, we now provide data verifying the robustness of the clustering analysis.

Rats were trained in a virtual reality behavioral task during which they were exposed to a pheromonal stimulus or neutral olfactory stimulus. LFP were simultaneously recorded in the PMCo and CA1 during the task. Both regions displayed theta oscillation and it is mentioned in the text that 35% of the time the two regions are coupled.

- How is this coupling related to the task (more concentrated during S3 ? Even in C4S3 ?).

Authors' response: We did not observe an increase in LFP coupling in sector 3 – corridor 3 (or in corridor 4 – sector 3). To further analyze the PMCo and CA1 interdependence during virtual navigation, in the new version of the manuscript we have incorporated the analysis of coherence and Granger causality of the signals recorded in these structures. The coherence allows us to detect temporal correlations between neural populations. In contrast to cross-correlation, used in the previous version of the article, we now use coherence, in short, the cross-spectrum between two signals normalized by the product of the auto-spectra (Jarvis and Mitra 2001). In this sense, coherence is generally considered to be a measure of consistency in phase differences and amplitude covariations between two signals.

The results are shown in a new Figure 2. Briefly, coherence shows a peak in the theta range. Thus, we found a coactivity between PMCo and CA1 in the theta range as a sign of exploratory behavior, a well-recognizable pattern in the CA1 area. However, our intention has also been to explore the directionality of this correlation, to check the origin and destination of this joint activity. Significant causality was detected in this frequency both in the PMCo→CA1 directionality and in the top-down directionality (CA1→PMCo). Bidirectional causality was also frequently found. Thus, there is a flow of information between the vomeronasal amygdala and the hippocampus during virtual navigation. The analysis of the causality in the PMCo→CA1 direction revealed a significant effect of the sector, due mainly to an increase between sectors 2 and 3. However, this effect is global, not specific of the presence of urinary stimuli.

With the addition of the coherence and causal directionality assessment, we believe that the results improve considerably, providing interesting data to verify the correlated neuronal activity between the two signals.

They used a method from Lopes-dos-Santos et al. (which was applied also to recordings in mice CA1) to perform single-cycle analysis of gamma/theta interaction in both the PMCo and the CA1 hippocampus. This analysis is very interesting and allowed to extract several spectral components in individual theta cycles in the present study, therefore contributing to validate the Lopes-Dos-Santos et al.'s analysis. In addition it shows that there are theta-nested gamma components in Amygdala and that a tSC5 fast gamma component in both Amygdala and hippocampus which is maintained across days 1 and 2.

- I wonder to what extent is the tSC5 specific to vomeronasal stimulation as it was also predominant when a neutral stimulus was applied? (C2 in day 2).

Authors' response: The reviewer is right in pointing out that the tSC5 component is not specific to the presence of vomeronasal stimuli, since it is also predominant when a neutral olfactory stimulus was applied (corridor 2 - sector 3). The unsupervised classification system shows that the whole pattern of

theta-nested gamma components is similar in PMCo and CA1 in the presence of vomeronasal stimuli, but not so much in the presence of olfactory stimuli. In this regard, it is now explained in the Results that when the neutral olfactory stimulus (sesame oil) was applied, the pattern of nested components in PMCo and CA1 did not cluster together, and a moderately strong tSC4 accompanies the tSC5 component. In contrast, in corridor 3-sector 3 (male urine) the strong tSC5 is accompanied by a moderate tSC1.

In any case, the absence of tSC5 elicited by new visual stimuli (corridor 2 – sector 2), and its presence elicited by chemosensory stimuli, suggests that the tSC5 reflects activity related to chemosensory exploration.

- It is argued that the shift from tSC5 to tSC4 during C4S3 reflects the spatial component. Could this shift be simply due to an absence of vomeronasal/olfactory stimulation?

Authors' response: our suggestion that the shift from tSC5 to tSC4 reflects the memory of the presence of vomeronasal stimuli in this same sector (Sector 3) in the previous corridor is certainly only an interpretation, and of course it is not the only possibility. The reviewer is right in pointing out the absence of chemosensory stimuli as a possible cause of the shift from tSC5 to tSC4. However, the absence of chemosensory stimuli would return the pattern of activity to that observed in sectors 1 or 4 (only visual stimuli from the walls present). However, this is not the case. The pattern of nested components is still relatively similar to that observed in corridor3-sector3, when the vomeronasal stimuli are presented, but with a tSC5 with less strength.

- It appears intriguing that the tSC5 to tSC4 shift was seen in the hippocampus but not in the PMCo which is expected to be modulated by changes in vomeronasal inputs.

Authors' response: We agree with the reviewer in that this observation is intriguing. Our interpretation is that the memory of the presence of vomeronasal stimuli in sector 3 is recovered from the pattern of activity in CA1. The tSC4 also appears in the PMCo in corridor4-sector3, but without losing the tSC5. A possible explanation would be a top-down control of PMCo from CA1. However, we find this suggestion too speculative and have not included it in the text.

I have no particular comment for the LTP experiments that convincingly show LTP in the PMCo, a novel result, and indicate that vomeronasal stimulation by male urine induced LTP in PMCo and CA1.

P16. Methods section, the “final location of the recording electrode...” should be “stimulation electrode” ?

Authors' response: yes, it should be “stimulation electrode”. We thank the reviewer for spotting this error, which we have corrected in the text.

Reviewer #2 (Remarks to the Author):

This is an interesting paper that makes an original contribution to the literature in both the fields of hippocampal learning and chemosensation. These findings suggest the importance of a pathway from vomeronasal system to hippocampal associative learning circuits, via the posteromedial cortical amygdala and entorhinal cortex. They speculate that activity in this pathway underlies the relational learning of spatial and identity information that underlies mouse territorial behaviour. A strong point of the study is the use of natural chemosignals present in urine, which have previously been shown to act as unconditioned stimuli for chemosensory learning. The fact that at least some of the mice appeared to show

long term potentiation of synaptic transmission in the posteromedial cortical amygdala and in CA1 of the hippocampus is a novel and highly interesting finding. It is consistent with the changes in molecular markers of plasticity.

However, there does appear to be a weakness in the way in which the LTP data have been analysed. Thirteen male mice were used for electrically induced LTP and 15 females for urine induced LTP. But there are only 6 to 8 datapoints in each of the figures. Individual animals contribute to the data points in each figure, but it is not clear whether the data points in different figures relate to the same animals. The authors state in the methods “Each case was classified as LTP if the amplitude value, obtained in the 20-25 min period after tetanic stimulation, was above the first quartile in the boxplot distribution. For each one of the selected cases... were compared against the basal values...”. Unless I’m mistaken, this implies that the authors have selected cases that show LTP and then analysed whether the selected cases show LTP in comparison with baseline values and in comparison with the 100% amplitude level. This would invalidate the statistical analysis, as the data have been selected and not randomly sampled. If this is the case it would render these statistics meaningless. Instead, I think that the authors need to be clearer on the proportion of the animals that showed LTP in each condition and maybe show results from all animals.

Authors’ response: We thank Dr Brennan for his insightful review and his constructive comments and suggestions, and of course for signing the review. Related to the way in which the LTP data are analyzed, we have followed the reviewer’s advice and we have now included in the ANOVA analysis all cases that were not discarded according to the histological data. Some additional cases were discarded due to loss of signal at some point during recording or by artifactual noise distorting the signal, which made it impossible to measure the evoked potential accurately. Thus, our analyses now include strong and weak LTP cases, showing therefore more variability (see new Figure 4, please note that some animals were used for LTP in only one of the structures). The results do not substantially change with respect to the previous version of the manuscript, with significant potentiation observed in the PMCo and CA1 either after tetanic stimulation of the accessory olfactory tract or induced with male urine. Additional LTP experiments have been performed in female mice with experimentally-induced anosmia (new Figure 5), to discard the role of the olfactory system in the synaptic potentiation induced by urine stimuli, as suggested by Reviewer #3 and explained in more detail below. The results of these additional experiments substantially strengthen the previous results showing LTP in the PMCo and CA1 induced by tetanic and urinary stimulation.

I’ve spent some time trying to understand the utility of the clustering analysis shown in figure 1. But I must admit that I find it very confusing and it doesn’t contain any indication of error. Wouldn’t it be simpler to replace it or supplement it with strength changes in the tSCs just for sector 3 of the different runways, along with an estimate of error or confidence?

Authors’ response: With regard to the utility of the clustering analysis of the tSCs, we show now in the new Figure 3 larger magnifications of the strength changes in the tSCs in corridor 2-sector 2 (evoked by new visual stimuli), corridor 2/sector 3 (evoked by sesame oil, a new olfactory stimulus), and corridor 3/sector 3 (evoked by male urine, a mixed olfactory-vomeronasal stimulus). As explained in response to

reviewer 1, we have added the cophenetic index (cophenetic correlation), a measure of the goodness of fit for the hierarchical clustering, similar to the coefficient of regression. Moreover, we now have applied a bootstrap analysis to further validate the results (x1000), a numerical approach to generating confidence intervals that use resampled data to estimate the sampling distribution of the maximum likelihood parameter estimates. In addition, we now detail in the text the analysis of the pattern of theta-nested gamma components in these sectors of the virtual reality, which are relevant to understand the activity of the PMCo and CA1 in the different contexts.

Other points to consider include:

In the abstract the authors claim to find “...synchronic activity in the vomeronasal amygdala and the dorsal CA1 of the hippocampus” But they need to be more specific about what was synchronised, as it would not be clear to a reader what this “synchronic activity” means.

Authors’ response: This relates to the second specific comment of Reviewer 1, which asks for more information about the synchronic activity in the vomeronasal amygdala and the dorsal CA1. We have performed new analysis on this synchronic activity, including the analysis of coherence and causality, as explained above in the reply to Reviewer #1 (please see Authors’ response #3).

Line 109 “...this synaptic plasticity is relayed to the dorsal hippocampus,..” I don’t think that you can say that synaptic plasticity is relayed between these brain areas. The evidence suggests that exposure to urine induces synaptic plasticity in both regions, but doesn’t imply transfer of plasticity.

Authors’ response: We agree with this observation and have corrected the text accordingly.

Line 275, Figure 2 legend, the authors state “induced by the tetanic stimulation of the accessory olfactory tract (A, B) and urinary pheromones (C, D)” but it would be better to refer to this as urinary stimulus (C, D), rather than pheromones, as pheromones not used specifically. The time axis for the boxplots is also unintelligible in figure 2.

Authors’ response: We agree with the Reviewer and have corrected the legend to refer to urinary stimulus. We apologize for the unintelligible letters in the time axis and thank the reviewer for spotting this error. This figure, which corresponds to the new Fig 4, has been amended.

Line 120 “and lower levels of activated GSK3 β in CA1 in urine-exposed animals than in controls (Fig. 3B; Extended Data Table 2).” The authors could make it clearer earlier on in the text that ratio of phosphorylated to non-phosphorylated GSK3beta is increased, which implies a decrease in active GSK3beta.

Authors’ response: We agree with this observation and have corrected the text accordingly (Results, lines 232-233, and Discussion, lines 363-368).

In Fig 3, it looks as though the authors have made a composite image of the gels with bands cut out of the original gel image. Would it not be better to show the original image for the whole gel?

Authors' response: We agree that it is better to show the whole gel and do so in the new Figure 6. In this regard, we ran a new experiment including a group of animals stimulated with citralva, so that we have now three experimental groups (saline, n=6; citralva, n=6; urine, n=6). For each target protein, Western blots were run in two gels, a larger one with 12 lanes and a smaller one with 7 lanes (sample in lane #1 was repeated in both gels for normalization of the results). The large gel corresponding to the hippocampal samples is shown in the new Fig. 6, and the small gel of the hippocampal samples and those corresponding to the amygdala (which yielded negative results) are shown in a new Supplementary Fig. 2. Please see also Authors Response below to the X comment of Reviewer#3.

In Fig 4a, it would be easier to assess co-localisation if there was a higher magnification insert in the figure.

Authors' response: We agree with the reviewer and have incorporated this higher magnification inset (new Fig. 7A).

The methods should really state which phase of the light dark cycle experiments were performed.

Authors' response: We now state in the Methods that all experiments were performed during the light phase, almost always during the morning.

I also don't find it clear what happens in the training corridor, is there just an absence of any stimulus so the animals are just trained to run?

Authors' response: To clarify the conditions of the training corridor, we have added a new Figure 1, showing the configuration of the four corridors. We now explain in the Methods that the training corridor was also divided in four sectors with visual cues in the walls different from those used in the testing corridors. In the sector 3 of the training corridor, a clean cotton swab was also presented, to habituate the animal to this object.

Were the order of testing corridors counterbalanced across individuals or always run in the same order?

Authors' response: We now specify in the Methods that the order of testing corridors was always the same, aiming to present in sector 3 first the neutral olfactory stimulus (corridor 2), then the urinary stimulus (corridor 3) and finally only the context (corridor 4).

Line 418, In the methods, the authors state that recording electrodes implanted in the AOB but then there is no mention of why this was done or what was found.

Authors' response: We implanted also a recording electrode in the accessory olfactory bulb in the first experimental animals, to guide the final location of the stimulating electrode in the aot. The evoked potential recorded by this electrode was only used for the location of the stimulating electrode in the aot, and was not recorded during the LTP experiments.

Line 420, "The final location of the recording electrode in aot was determined guided by the maximum response evoked in the accessory olfactory bulb" I think this should be stimulating electrode, as there was no recording electrode in the aot.

Authors' response: yes, it should be “stimulation electrode”. We thank the reviewer for spotting this error, which we have corrected in the text.

Line 452, “For statistical analysis, the fEPSP values were grouped in sets of 5 measures and averaged,...” It’s not clear what’s meant by “grouped in sets of 5 measures”. Does this mean 5 neighbouring timepoints were averaged?

Authors' response: In the previous version, the data of the 5 neighboring timepoints were averaged for analysis and plotting. However, in this new version we use the average of the 6 neighboring timepoints, so that each average represents one minute (the stimulation pulses were delivered 1 every 10 seconds, and thus 6 per minute). This is detailed in the Methods (Section “Long-term potentiation methods”, lines 553-555).

Line 464, The authors state that they collected histological sections from the LTP animals. But they don’t say any more about why they were collected or how they were used. Were they analysed for electrode placement? Could this have been used to select animals to analyse rather than the presence of LTP?

Authors' response: Histological sections from the animals used in the LTP experiments were examined for electrode placement. A new supplementary Figure (supplementary Fig 1) has been added to show representative electrode placements.

Reviewer #3 (Remarks to the Author):

The manuscript from Villafranca-Faus et al. reports a new interesting view that correlates chemosensory-induced activity in the cortical amygdala and parallel activation of the dorsal CA1 of the hippocampus, likely related to spatial learning. Although experiments are well conducted, conclusions are often not fully supported by provided data, in particular claims of pheromone influence on hippocampal learning. I find the work lacks some critical controls.

Major issues:

1. The link with the vomeronasal system is based mostly on circumstantial evidence. The PMCo largely receive vomeronasal inputs, but it cannot be excluded that other sensory information (i.e. olfactory) participate in modulating hippocampal activity, even from the same location. The same for the LEnt. Use of a neutral olfactory stimulus (sesame oil) in Fig. 1 goes in the right direction, but is not robust enough. To better assess pheromone and vomeronasal roles, it would be important to include a control with a mouse model deficient for vomeronasal function (genetic or surgical VNO ablation). If the authors’ hypothesis is correct, a VNO-deficient mouse would retain full response after tetanic stimulation in the paradigm of Fig 2, but responses after urine stimulation will be lost in both the aot and hippocampus. Some level of activity still present at any of these two levels would be indicative of partial involvement of the main olfactory system and not only the vomeronasal system. Shown activity after application of urine rather suggest a role of the main olfactory system as VNO pumping is likely not active in anesthetized animals. Using purified pheromones instead of a complex mixture like urine may also elucidate this point.

Authors' response: We agree with Reviewer #3 in the need for additional controls to discard a role of the olfactory system in both the LTP experiments and the experiments aiming to investigate the expression of proteins related to synaptic plasticity. With regard to the LTP experiments, Reviewer #3 suggests the possibility to use a mouse model deficient for vomeronasal function (genetic or surgical VNO ablation). However, that model would be useful to demonstrate the involvement of the olfactory system, which is not our objective in this study. We find it quite likely that olfactory stimuli can induce LTP in the hippocampus and maybe also in the chemosensory amygdala. Thus, since we aim to demonstrate the involvement of the vomeronasal system, we run the LTP experiments in animals rendered anosmic employing zinc sulfate irrigation of the nasal mucosa. The successful generation of anosmia was tested in habituation-dishabituation tests. In this experimental model, we have tested the induction of LTP in the PMCo and the hippocampal CA1 using male urine and a neutral olfactory stimulus such as the odorant citralva. The results show successful LTP induction with urine and no effect of the odorant citralva. These results are described in the new manuscript at the end of the section of the LTP induction with urine (lines 191-215), and illustrated in a new Fig. 5.

Related to this point, Reviewer 3 suggests that VNO pumping may not be active in anesthetized animals. To address this issue, following Ben-Shaul et al (2010, PNAS, doi:10.1073/pnas.0915147107), at the end of the LTP experiments we delivered the fluorescent dye Dil in the nasal cavity ipsilateral to the stimulating and recording electrodes, in the same way as we deliver the urine during the LTP induction. Following perfusion, we obtained histological sections of the VNO. We could observe the presence of intense dye labeling in the lumen of the organ, as well as in the sensory epithelium (described in Results, lines 195-197; illustrated in new Fig. 5B). This result shows that, either by diffusion or active pumping, the liquid (urine or dye) delivered into the nostril can reach the sensory epithelium of the VNO. Regarding the possibility of active pumping, this mechanism is under sympathetic control (Ben-Shaul et al, 2010, PNAS). A previous report suggests that urethane anesthesia is likely not decreasing the sympathetic activity (Shimokawa et al., 1998, Differential effects of anesthetics on sympathetic nerve activity and arterial baroreceptor reflex in chronically instrumented rats. J Auton Nerv Syst. doi: 10.1016/s0165-1838(98)00084-8).

We would like to insist that we are not suggesting that the olfactory system is not involved in learning induced by urinary cues. Actually, an active role of the olfactory systems is a very likely hypothesis, and we have stated it in the Discussion.

2. In Fig. 3, the expression analysis of proteins related to synaptic plasticity in the amygdala and dorsal hippocampus is not clear to me and needs further validation. First, using an olfactory neutral (non-pheromonal) stimulus to demonstrate that olfactory stimulation alone is not sufficient to increase expression. Confirmation of the results using a second alternative method (qRT-PCR, etc.) would considerably support the conclusions.

Authors' response: Following Reviewer#3 suggestion, we have repeated the experiment with the addition of a group of animals exposed to citralva (an olfactory neutral, non-pheromonal stimulus) to investigate the effects of olfactory stimulation. The results do not change significantly in the hippocampus, and the animals exposed to citralva did not show an increased expression of pAkt and pGSK3 β . Thus, in the case of

the hippocampus the addition of the group exposed to citralva confirmed our previous results suggesting that the increased expression of pAkt and pGSK3 β was related to the synaptic plasticity induced by the urinary stimuli. In contrast, the results do change in the amygdala respect to the first version, and the group exposed to citralva showed an expression of pCREB and CRCT1 that was not statistically different from that observed in the group exposed to urine. Thus, we have modified the results accordingly and we now show the results of the amygdala in a new Supplementary Fig. 2.

Since three out of four of the markers of synaptic plasticity are phosphorylated proteins (pCREB, pAKT and pGSK3), it was not possible to confirm these results using qPCR.

3. Fig. 4 lacks measurements or quantification data. Do the authors find other overlapping brain areas (apart from dLEnt) of retrogradely labeled cells and anterogradely labeled fibers? At what extent? What proportion of cells from the dLEnt receives innervation from the PMCo? How many of those send projections to the hippocampus?

Authors' response: We carefully looked for other possible areas (apart from dLEnt) of retrogradely labeled cells and anterogradely labeled fibers. A meticulous analysis of the cases of injections of the retrograde tracer Fluorogold in CA1 and the anterograde tracer BDA in the PMCo showed virtually no other areas of overlapping. A few retrogradely labelled neurons appeared in the medial amygdaloid nucleus, which receives projections from the PMCo (and the AOB). However, labelling was really very scarce and not present in every case. In addition, we could not confirm the projection from the medial amygdala to the dorsal hippocampus using anterograde tracing (Pardo-Bellver et al., 2012, *Front Neuroanat*, doi: 10.3389/fnana.2012.00033). In conclusion, the combined tract-tracing study did not reveal any other area that could relay information from the PMCo (or other vomeronasal structure) to the dorsal CA1.

Reviewer #3 also asks for quantitative data on the tract-tracing studies, particularly regarding the proportion of cells from the dLEnt receiving innervation from the PMCo and projecting to the hippocampus. Unfortunately, the tract-tracing techniques do not allow a reliable quantification, because the results will always depend on the size and exact location of the injection site (in our experiments of the two injection sites, i.e., Fluorogold in CA1 and BDA in PMCo). The number of retrogradely labeled cells with FG will therefore vary depending on how much tracer is injected and captured by the axon terminals of the entorhinal cells, and in the same way the number of anterogradely labeled axons will depend on how much tracer is captured and transported by the PMCo neurons. Thus, we can only qualitatively describe the juxtaposition of BDA labeled fibers and Fluorogold positive cells. In any case, from a qualitative point of view, it appears that most of the cells in the dorsolateral aspect of Ent layer II are retrogradely labeled with Fluorogold. To give a similar qualitative observation of the projection from the PMCo to the dLEnt, we injected Fluorogold in the dLEnt and evaluate the retrograde labeling in the PMCo. The results show that many PMCo neurons were retrogradely labeled, especially in the caudal two thirds of this nucleus, and located preferentially in the medial aspect of the PMCo. These results are included in the text and illustrated in a new supplementary Figure 3.

4. c-Fos expression in the dLEnt in females exposed to male pheromones is an important result. It would be helpful to show representative images as well as raw measurements of stimulated vs controls, and not only a correlation chart. Are these cFos+ cells in the dLEnt also positive for reelin?

Authors' response: Following the Reviewers' advice for the experiments of LTP and molecular markers of synaptic plasticity, we have also repeated the experiment investigating c-Fos expression using a neutral olfactory stimulus (citrulva) to compare the odorant-induced c-Fos with the urine-induced c-Fos, and also to investigate, as suggested by the reviewer, whether c-Fos-positive cells in the dLEnt are also positive for reelin. The results replicate the previous results (which used clean bedding as control) in the PMCo and dLEnt, and show that the Fos-positive cells also expressed reelin. However, in this case we found no difference in the number of activated cells in CA1. We have added these results to the manuscript. and suggest in the Discussion that urine-induced c-Fos expression in the PMCo and the reelin-positive cells in dLEnt (compared to the c-Fos induced by citrulva) is due to the vomeronasal cues present in urine. In contrast, the activation of the CA1 pyramidal cells is not different between the animals exposed to citrulva and those exposed to urine. We found a differential expression of c-Fos in CA1 in the previous experiment (reanalyzing the preparations of Moncho-Bogani et al., 2005) because the control animals were exposed to clean bedding, whereas the experimental group was exposed to male-soiled bedding in a particular corner of the test cage. These results are illustrated in a new Figure (Fig. 8), including representative images and the boxplot of raw measurements, as suggested by the reviewer.

Minor:

A scheme of the experimental preparation of Fig. 1 with the head-fixed mice, the different virtual navigation corridors and sequential stimuli will help to understand an otherwise complex figure.

Authors' response: we agree with the Reviewer and have prepared a new Figure 1 showing the characteristics of the virtual reality, and include a supplementary video of the virtual navigation.

We are very grateful to all three reviewers by their constructive and helpful comments. We sincerely believed that the modified manuscript, thanks to the reviewers' comments and suggestions, is notably improved and the results now support more in a more solid way the proposed interpretation.

REVIEWERS' COMMENTS

Reviewer #1 (Remarks to the Author):

The present manuscript is a revised version of an initial submission in september/october 2020. The initial manuscript has been substantially amended and the revision includes additional experiments and analyses that improve the paper and support more strongly the hypothesis that vomeronal signals are possibly incorporated in the hippocampal spatial map. It is convincingly shown that pheromonal information induces correlated activity and plasticity in the PMCo part of the amygdala and the CA1 of the hippocampus and that this synchrony is subtended by a PMCo-LEnt-CA1 pathway. The authors have satisfactorily responded to my comments and overall they have made a considerable effort to respond all the reviewer's comments. I therefore recommend this manuscript to be published.

Reviewer #2 (Remarks to the Author):

I'm happy that the authors have satisfactorily responded to all of my previous comments. The extra evidence included by the authors has greatly strengthened the paper, which is a high quality and highly interesting body of work. I have spotted, what look like a couple of minor typographical omissions.

Line 98 " $F_{3} = 3.71, p = 0.015$), with the increase of ratio with directionality PMCo → CA1 in the transition from s2 to s3 ($s2-s3, t_{df} = 2.81, p = 0.031$)." - it appears that there might be a degree of freedom missing in the original F ratio " F_{3} " It also looks as though degrees of freedom are missing for the t statistic " t_{df} ".

Another minor point, which doesn't affect the conclusions and doesn't require a change is that the authors should be careful in suggesting that sesame oil is a neutral stimulus. Unlike citralva, which they've convincingly demonstrated is a neutral stimulus, sesame oil is a naturally occurring foodstuff. It's well documented that mice have an innate preference for peanut odour, so it would not be surprising if they also had an innate preference for sesame oil. It doesn't affect the findings, but something to consider going forwards.

Peter Brennan

Reviewer #3 (Remarks to the Author):

The revised manuscript incorporates new data and adequately addresses all of my initial concerns. The new figures greatly improved the paper.

POINT BY POINT REPLY TO REVIEWERS

Reviewer #1 (Remarks to the Author):

The present manuscript is a revised version of an initial submission in september/october 2020. The initial manuscript has been substantially amended and the revision includes additional experiments and analyses that improve the paper and support more strongly the hypothesis that vomeronal signals are possibly incorporated in the hippocampal spatial map. It is convincingly shown that pheromonal information induces correlated activity and plasticity in the PMCo part of the amygdala and the CA1 of the hippocampus and that this synchrony is subtended by a PMCo-LEnt-CA1 pathway. The authors have satisfactorily responded to my comments and overall they have made a considerable effort to respond all the reviewer's comments. I therefore recommend this manuscript to be published.

Authors' response: We thank the reviewer for these comments.

Reviewer #2 (Remarks to the Author):

I'm happy that the authors have satisfactorily responded to all of my previous comments. The extra evidence included by the authors has greatly strengthened the paper, which is a high quality and highly interesting body of work.

Authors' response: We thank Dr Brennan for his insightful review and his constructive comments and suggestions.

I have spotted, what look like a couple of minor typographical omissions Line 98 "(F3 = 3.71, p = 0.015), with the increase of ratio with directionality PMCo → CA1 in the transition from s2 to s3 (s2-s3, tdf = 2.81, p = 0.031)." - it appears that there might be a degree of freedom missing in the original F ratio "F3" It also looks as though degrees of freedom are missing for the t statistic "tdf".

Authors' response: We thank the reviewer for spotting these error, which have been corrected.

Another minor point, which doesn't affect the conclusions and doesn't require a change is that the authors should be careful in suggesting that sesame oil is a neutral stimulus. Unlike citralva, which they've convincingly demonstrated is a neutral stimulus, sesame oil is a naturally occurring foodstuff. It's well documented that mice have an innate preference for peanut odour, so it would not be surprising if they

carrer **del Doctor Moliner, 50 ; BURJASSOT 46100; SPAIN**

telèfon **(34) 96 354 3383**

e-mail **Enrique.Lanuza@uv.es**

also had an innate preference for sesame oil. It doesn't affect the findings, but something to consider going forwards.

Authors' response: We agree with the reviewer in that sesame oil might not be an olfactory neutral stimulus, and we have modified the text to refer to sesame oil as a “non-pheromonal olfactory stimulus”.

Reviewer #3 (Remarks to the Author):

The revised manuscript incorporates new data and adequately addresses all of my initial concerns. The new figures greatly improved the paper. Major issues:

Authors' response: We thank the reviewer for these comments.

We are very grateful to all three reviewers by their constructive and helpful comments along the review process. We would like to acknowledge the improvement of the work with the revision thanks to the Reviewers' suggestions.